# FiT: Parameter Efficient Few-shot Transfer Learning for Personalized and Federated Image Classification

**Aliaksandra Shysheya**[*]
University of Cambridge
as2975@cam.ac.uk

**John Bronskill**[*]
University of Cambridge
jfb54@cam.ac.uk

**Massimiliano Patacchiola**
University of Cambridge
mp2008@cam.ac.uk

**Sebastian Nowozin**[†]
nowozin@gmail.com

**Richard E. Turner**
University of Cambridge
ret26@cam.ac.uk

## Abstract

Modern deep learning systems are increasingly deployed in situations such as personalization and federated learning where it is necessary to support i) learning on small amounts of data, and ii) communication efficient distributed training protocols. In this work, we develop FiLM Transfer (FiT) which fulfills these requirements in the image classification setting by combining ideas from transfer learning (fixed pretrained backbones and fine-tuned FiLM adapter layers) and meta-learning (automatically configured Naive Bayes classifiers and episodic training) to yield parameter efficient models with superior classification accuracy at low-shot. The resulting parameter efficiency is key for enabling few-shot learning, inexpensive model updates for personalization, and communication efficient federated learning. We experiment with FiT on a wide range of downstream datasets and show that it achieves better classification accuracy than the leading Big Transfer (BiT) algorithm at low-shot and achieves state-of-the art accuracy on the challenging VTAB-1k benchmark, with fewer than $1\%$ of the updateable parameters. Finally, we demonstrate the parameter efficiency and superior accuracy of FiT in distributed low-shot applications including model personalization and federated learning where model update size is an important performance metric.

## 1 Introduction

With the success of the commercial application of deep learning in many fields such as computer vision (Schroff et al., 2015), natural language processing (Brown et al., 2020), speech recognition (Xiong et al., 2018), and language translation (Wu et al., 2016), an increasing number of models are being trained on central servers and then deployed on remote devices, often to personalize a model to a specific user's needs. Personalization requires models that can be updated inexpensively by minimizing the number of parameters that need to be stored and / or transmitted and frequently calls for few-shot learning methods as the amount of training data from an individual user may be small (Massiceti et al., 2021). At the same time, for privacy, security, and performance reasons, it can be advantageous to use federated learning where a model is trained on an array of remote devices, each with different data, and share gradient or parameter updates instead of training data with a central server (McMahan et al., 2017). In the federated learning setting, in order to minimize communication cost with the server, it is also beneficial to have models with a small number of parameters that need to be updated for each training round conducted by remote clients. The amount of training data available to the clients is often small, again necessitating few-shot learning approaches.

In order to develop data-efficient and parameter-efficient learning systems, we draw on ideas developed by the few-shot learning community. Few-shot learning approaches can be characterized in

---

[*]Authors contributed equally.
[†]Work performed while at Microsoft Research.

Figure 1: **FIT is significantly more parameter efficient than BiT.** Results summary for FIT and BiT in classification, personalization, and federated learning scenarios using the BiT-M-R50x1 backbone. The *Parameters* plots refer to the typical number of updateable parameters in each model, while the *Cost* plot refers to the total client-server communication cost during federated training. In all settings, FIT achieves similar or better classification accuracy using orders of magnitude fewer updateable parameters and communication cost. Refer to Table 2, Table 4, and Table 5 for more detail.

terms of shared and updateable parameters. From a statistical perspective, shared parameters capture similarities between datasets, while updateable parameters capture the differences. Updateable parameters are those that are either recomputed or learned as the model is updated or retrained, whereas shared parameters are fixed. In personalized or federated settings, it is key to minimize the number of updateable parameters, while still retaining the capacity to adapt.

Broadly, there are two different approaches to few-shot learning: meta-learning (Hospedales et al., 2020) and transfer learning (fine-tuning) (Yosinski et al., 2014). Meta-learning approaches provide methods that have a small number of updatable parameters (Requeima et al., 2019). However, while meta-learners can perform strongly on datasets that are similar to those they are meta-trained on, their accuracy suffers when tested on datasets that are significantly different (Dumoulin et al., 2021). Transfer learning algorithms often outperform meta-learners, especially on diverse datasets and even at low-shot (Dumoulin et al., 2021; Tian et al., 2020). However, the leading Big Transfer (BiT) (Dumoulin et al., 2021; Kolesnikov et al., 2019) algorithm requires every parameter in a large network to be updated. In summary, performant transfer learners are parameter-inefficient, and parameter-efficient few-shot learners perform relatively poorly.

In this work we propose FiLM Transfer or FiT, a novel method that synthesizes ideas from both the transfer learning and meta-learning communities in order to achieve the best of both worlds – parameter efficiency without sacrificing accuracy, even when there are only a small number of training examples available. From transfer learning, we take advantage of backbones pretrained on large image datasets and the use of fine-tuned parameter efficient adapters. From meta-learning, we take advantage of metric learning based final layer classifiers trained with episodic protocols that we show are more effective than the conventional linear layer classifier.

We experiment with FIT on a wide range of downstream datasets and show that it achieves better classification accuracy at low-shot with two orders of magnitude fewer updateable parameters when compared to BiT (Kolesnikov et al., 2019) and competitive accuracy when more data are available. We also demonstrate the benefits of FIT on a low-shot real-world model personalization application and in a demanding few-shot federated learning scenario. A summary of our results is shown in Fig. 1, where we see that FIT has superior parameter efficiency and classification accuracy compared to BiT in multiple settings. Our contributions:

- A parameter and data efficient network architecture for low-shot transfer learning that (i) utilizes frozen backbones pretrained on large image datasets; (ii) augments the backbone with parameter efficient FiLM (Perez et al., 2018) layers in order to adapt to a new task; and (iii) makes novel use of an automatically configured Naive Bayes final layer classifier instead of the usual linear layer, saving a large number of updateable parameters, yet improving classification performance;
- A meta-learning inspired episodic training protocol for low-shot fine-tuning requiring no data augmentation, no regularization, and a minimal set of hyper-parameters;
- Superior classification accuracy at low-shot on standard downstream datasets and state-of-the-art results on the challenging VTAB-1k benchmark (74.9% for backbones pretrained on ImageNet-21k) while using $\approx 1\%$ of the updateable parameters when compared to the leading method BiT;
- Demonstration of superior parameter efficiency and classification accuracy in distributed low-shot personalization and federated learning applications where model update size is a key performance metric. We show that the FIT communication cost is more than 3 orders of magnitude lower than BiT (7M versus 14B parameters transmitted) in our CIFAR100 federated learning experiment.

## 2 FiLM Transfer (FiT)

In this section we detail the FiT algorithm focusing on the few-shot image classification scenario.

**Preliminaries** We denote input images $\boldsymbol{x} \in \mathbb{R}^{ch \times W \times H}$ where $W$ is the width, $H$ the height, $ch$ the number of channels and image labels $y \in \{1, \ldots, C\}$ where $C$ is the number of image classes indexed by $c$. Assume that we have access to a model $f(\boldsymbol{x}) = h_{\boldsymbol{\phi}}(b_{\boldsymbol{\theta}}(\boldsymbol{x}))$ that outputs class-probabilities for an image $p(y = c|\boldsymbol{x}, \boldsymbol{\theta}, \boldsymbol{\phi})$ for $c = 1, \ldots, C$ and is comprised of a feature extractor backbone $b_{\boldsymbol{\theta}}(\boldsymbol{x}) \in \mathbb{R}^{d_b}$ with parameters $\theta$ that has been pretrained on a large upstream dataset such as Imagenet where $d_b$ is the output feature dimension and a final layer classifier or head $h_{\boldsymbol{\phi}}(\cdot) \in \mathbb{R}^C$ with weights $\boldsymbol{\phi}$. Let $\mathcal{D} = \{(\boldsymbol{x}_n, y_n)\}_{n=1}^N$ be the downstream dataset that we wish to fine-tune the model $f$ to.

**FiT Backbone** For the network backbone, we freeze the parameters $\theta$ to the values learned during upstream pretraining. To enable parameter-efficient and flexible adaptation of the backbone, we add Feature-wise Linear Modulation (FiLM) (Perez et al., 2018) layers with parameters $\boldsymbol{\psi}$ at strategic points within $b_{\boldsymbol{\theta}}$. A FiLM layer scales and shifts the activations $\boldsymbol{a}_{ij}$ arising from the $j^{th}$ channel of a convolutional layer in the $i^{th}$ block of the backbone as $\texttt{FiLM}(\boldsymbol{a}_{ij}, \gamma_{ij}, \beta_{ij}) = \gamma_{ij}\boldsymbol{a}_{ij} + \beta_{ij}$, where $\gamma_{ij}$ and $\beta_{ij}$ are scalars. The set of FiLM parameters $\boldsymbol{\psi} = \{\gamma_{ij}, \beta_{ij}\}$ is learned during fine-tuning. We add FiLM layers following the middle $3 \times 3$ convolutional layer in each ResNetV2 (He et al., 2016b) block and also one at the end of the backbone prior to the head. Fig. A.1a illustrates a FiLM layer operating on a convolutional layer, and Fig. A.1b illustrates how a FiLM layer can be added to a ResNetV2 network block. FiLM layers can be similarly added to other backbones. An advantage of FiLM layers is that they enable expressive feature adaptation while adding only a small number of parameters (Perez et al., 2018). For example, in a ResNet50 with a FiLM layer in every block, the set of FiLM parameters $\boldsymbol{\psi}$ account for only 11648 parameters which is fewer than 0.05% of the parameters in $b_{\boldsymbol{\theta}}$. We show in Section 4 that FiLM layers allow the model to adapt to a broad class of datasets. Fig. A.2a and Fig. A.2b show the magnitude of the FiLM parameters as a function of layer for FiT-LDA on CIFAR100 and SHVN, respectively. Refer to Appendix A.7 for additional detail.

**FiT Head** For the head of the network, we use a specially tailored Gaussian Naive Bayes classifier. Unlike a linear head, this head can be automatically configured directly from data and has only a small number of free parameters which must be learned, ideal for few-shot, personalization and federated learning. We will also show that this head is often more accurate than a standard linear head. The class probability for a test point $\boldsymbol{x}^*$ is:

$$p(y^* = c|b_{\boldsymbol{\theta}, \boldsymbol{\psi}}(\boldsymbol{x}^*), \boldsymbol{\pi}, \boldsymbol{\mu}, \boldsymbol{\Sigma}) = \frac{\pi_c \mathcal{N}(b_{\boldsymbol{\theta}, \boldsymbol{\psi}}(\boldsymbol{x}^*)|\mu_c, \Sigma_c))}{\sum_{c'}^C \pi_{c'} \mathcal{N}(b_{\boldsymbol{\theta}, \boldsymbol{\psi}}(\boldsymbol{x}^*)|\mu_{c'}, \Sigma_{c'})} \tag{1}$$

where $\pi_c = \frac{N_c}{N}$, $\mu_c = \frac{1}{N_c} \sum_{i=1}^{N_c} b_{\boldsymbol{\theta}, \boldsymbol{\psi}}(\boldsymbol{x}_i)$, $\Sigma_c = \frac{1}{N_c} \sum_{i=1}^{N_c} (b_{\boldsymbol{\theta}, \boldsymbol{\psi}}(\boldsymbol{x}_i) - \mu_c)(b_{\boldsymbol{\theta}, \boldsymbol{\psi}}(\boldsymbol{x}_i) - \mu_c)^T$

are the maximum likelihood estimates, $N_c$ is the number of examples of class $c$ in $\mathcal{D}$, and $\mathcal{N}(z|\mu, \Sigma)$ is a multivariate Gaussian over $z$ with mean $\mu$ and covariance $\Sigma$.

Estimating the mean $\mu_c$ for each class $c$ is straightforward and incurs a total storage cost of $Cd_b$. However, estimating the covariance $\Sigma_c$ for each class $c$ is challenging when the number of examples per class $N_c$ is small and the embedding dimension of the backbone $d_b$ is large. In addition, the storage cost for the covariance matrices may be prohibitively high if $d_b$ is large. Here, we use three different approximations to the covariance in place of $\Sigma_c$ in Eq. (1) (Fisher, 1936; Duda et al., 2012):

- **Quadratic Discriminant Analysis (QDA)**: $\Sigma_{\text{QDA}} = e_1\Sigma_{class} + e_2\Sigma_{task} + e_3\boldsymbol{I}$
- **Linear Discriminant Analysis (LDA)**: $\Sigma_{\text{LDA}} = e_2\Sigma_{task} + e_3\boldsymbol{I}$
- **ProtoNets** (Snell et al., 2017): $\Sigma_{\text{PN}} = \boldsymbol{I}$; i.e. there is no covariance and the class representation is parameterized only by $\mu_c$ and the classifier logits are formed by computing the squared Euclidean distance between the feature representation of a test point $b_{\boldsymbol{\theta}, \boldsymbol{\psi}}(\boldsymbol{x}^*)$ and each of the class means.

In the above, $\Sigma_{class}$ is the computed covariance of the examples in class $c$ in $\mathcal{D}$, $\Sigma_{task}$ is the computed covariance of all the examples in $\mathcal{D}$ assuming they arise from a single Gaussian with a single mean, $e = \{e_1, e_2, e_3\}$ are weights learned during training, and the identity matrix $\boldsymbol{I}$ is used as a regularizer. QDA mainly serves as a baseline since it has a very large set of updateable parameters arising from the fact that it stores a covariance matrix for each class in the dataset. LDA is far more parameter efficient than QDA, sharing a single covariance matrix across all classes. We show that LDA leads to

Table 1: Shared and updateable parameters for the transfer learning methods considered. The Example column contains the updateable parameters for all methods using a BiT-M-R50x1 backbone with $|\boldsymbol{\theta}| = 23{,}500{,}352$, $\boldsymbol{\psi} = 11{,}648$, $d_b = 2048$, and $C = 10$.

| Method | Shared | Updateable | Example |
|---|---|---|---|
| BiT | 0 | $|\boldsymbol{\theta}| + |\boldsymbol{\phi}| = |\boldsymbol{\theta}| + Cd_b$ | 23,520,832 |
| FIT - QDA | $|\boldsymbol{\theta}|$ | $|\boldsymbol{\psi}| + |\boldsymbol{\mu}| + |\boldsymbol{\Sigma}| + |e| = |\boldsymbol{\psi}| + Cd_b + C\frac{d_b(d_b+1)}{2} + 3$ | 21,013,891 |
| FIT - LDA | $|\boldsymbol{\theta}|$ | $|\boldsymbol{\psi}| + (|\boldsymbol{\mu}| + |\boldsymbol{\Sigma}|) + |e| = |\boldsymbol{\psi}| + C(d_b + 1) + 2$ | 32,140 |
| FIT - ProtoNets | $|\boldsymbol{\theta}|$ | $|\boldsymbol{\psi}| + |\boldsymbol{\mu}| = |\boldsymbol{\psi}| + Cd_b$ | 32,128 |

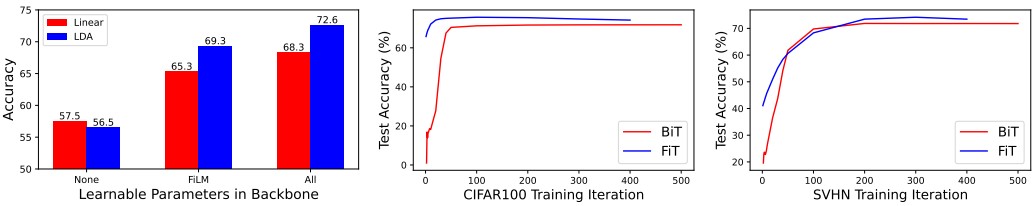

(a) LDA outperforms linear head.  (b) FIT-LDA converges more quickly than BiT.

Figure 2: (a) Average accuracy on VTAB-1k for linear and LDA heads versus learnable parameters in the backbone. (b) Test accuracy versus training iteration for CIFAR100 and SVHN on VTAB-1k.

very similar performance to QDA. The number of model shared and updateable parameters for the three FIT variants as well as the BiT algorithm are detailed in Table 1. Refer to Appendix A.2 for details on the parameter calculations.

An empirical justification for the use of the FIT-LDA head is shown in Fig. 2a where it outperforms a linear head in the case of FiLM and when all the backbone parameters are learned. Refer to Table A.5 for details. In Fig. 2b, we see for both datasets, FIT-LDA converges significantly faster than BiT, which uses a linear head. The primary limitation of the Naive Bayes head is the higher (versus linear) computational cost due to having to invert a $d_b \times d_b$ covariance matrix on each training iteration.

**FIT Training**  We learn the FiLM parameters $\boldsymbol{\psi}$ and the covariance weights $\boldsymbol{e}$ via fine-tuning (parameters $\boldsymbol{\theta}$ are fixed from pretraining). One approach would be to apply standard batch training on the downstream dataset, however it was hard to balance under- and over-fitting using this setup. Instead, we found that an approach inspired by *episodic training* (Vinyals et al., 2016) that is often used in meta-learning yielded better performance. We refer to this approach as *episodic fine tuning* and it works as follows. Note that we require 'training' data to compute $\boldsymbol{\pi}$, $\boldsymbol{\mu}$, $\boldsymbol{\Sigma}$ to configure the head, and a 'test' set to optimize $\boldsymbol{\psi}$ and $\boldsymbol{e}$ via gradient ascent. Thus, from the downstream dataset $\mathcal{D}$, we derive two sets – $\mathcal{D}_{train}$ and $\mathcal{D}_{test}$. If $\mathcal{D}$ is sufficiently large ($|\mathcal{D}| \approx 1000$), such that overfitting is not an issue, we set $\mathcal{D}_{train} = \mathcal{D}_{test} = \mathcal{D}$. Otherwise, in the few-shot scenario, we randomly split $\mathcal{D}$ into $\mathcal{D}_{train}$ and $\mathcal{D}_{test}$ such that the number of examples or *shots* in each class $c$ are roughy equal in both partitions and that there is at least one example of each class in both. Refer to Algorithm A.1 for details. For each training iteration, we sample a task $\tau$ consisting of a *support* set $\mathcal{D}_S^\tau$ drawn from $\mathcal{D}_{train}$ with $S_\tau$ examples and a *query* $\mathcal{D}_Q^\tau$ set drawn from $\mathcal{D}_{test}$ with $Q_\tau$ examples. First, $\mathcal{D}_S^\tau$ is formed by randomly choosing a subset of classes selected from the range of available classes in $\mathcal{D}_{train}$. Second, the number of shots to use for each selected class is randomly selected from the range of available examples in each class of $\mathcal{D}_{train}$ with the goal of keeping the examples per class as equal as possible. Third, $\mathcal{D}_Q^\tau$ is formed by using the classes selected for $\mathcal{D}_S^\tau$ and all available examples from $\mathcal{D}_{test}$ in those classes up to a limit of 2000 examples. Refer to Algorithm A.2 for details. Episodic fine-tuning is crucial to achieving the best classification accuracy with the Naive Bayes head. If all of $\mathcal{D}_{train}$ and $\mathcal{D}_{test}$ are used for every iteration, overfitting occurs, limiting accuracy (see Table A.2 and Table A.4). The support set $\mathcal{D}_S^\tau$ is then used to compute $\boldsymbol{\pi}$, $\boldsymbol{\mu}$, and $\boldsymbol{\Sigma}$ and we then use $\mathcal{D}_Q = \{\{\boldsymbol{x}_q^{\tau*}, y_q^{\tau*}\}_{q=1}^{Q_\tau}\}_{\tau=1}^T$ to train $\boldsymbol{\psi}$ and $\boldsymbol{e}$ with maximum likelihood. We optimize the following:

$$\hat{\mathcal{L}}(\boldsymbol{\psi}, \boldsymbol{e}) = \sum_{\tau=1}^T \sum_{q=1}^{Q_\tau} \log p\left(y_q^{\tau*} | h_e(b_{\theta,\psi}(\boldsymbol{x}_q^{\tau*})), \boldsymbol{\pi}(\mathcal{D}_s^\tau), \boldsymbol{\mu}(\mathcal{D}_s^\tau), \boldsymbol{\Sigma}(\mathcal{D}_s^\tau)\right). \quad (2)$$

FIT training hyper-parameters include a learning rate, $|\mathcal{D}_S^\tau|$, and the number of training iterations. For the transfer learning experiments in Section 4 these are set to constant values across all datasets and do not need to be tuned based on a validation set. We do not augment the training data. In the 1-shot case, we do not perform episodic fine-tuning and leave the FiLM parameters at their initial value of $\boldsymbol{\gamma} = 1, \boldsymbol{\beta} = 0$ and $\boldsymbol{e} = (0.5, 0.5, 1.0)$ and predict as described next. In Section 4 we show this can yield results that better those when augmentation and training steps are taken on 1-shot data.

**FIT Prediction** Once the FiLM parameters $\psi$ and covariance weights $\boldsymbol{e}$ have been learned, we use $\mathcal{D}$ for the support set to compute $\pi_c$, $\mu_c$, and $\Sigma_c$ for each class $c$ and then Eq. (1) can be used to make a prediction for any unseen test input.

## 3 RELATED WORK

We take inspiration from residual adapters (Rebuffi et al., 2017; 2018) where parameter efficient adapters are inserted into a ResNet with frozen pretrained weights. The adapter parameters and the final layer linear classifier are then learned via fine-tuning. More recently, a myriad of additional parameter efficient adapters have been proposed including FiLM, Adapter (Houlsby et al., 2019), LoRA (Hu et al., 2021), VPT (Jia et al., 2022), AdaptFormer (Chen et al., 2022), NOAH (Zhang et al., 2022), Convpass (Jie & Deng, 2022), (Mudrakarta et al., 2019), and CaSE (Patacchiola et al., 2022). For FIT we use FiLM as it is the most parameter efficient adapter, yet it allows for expressive adaptation, and can be used in various backbone architectures including ConvNets and Transformers.

To date, transfer learning systems that employ adapters use a linear head for the final classification layer. In meta-learning systems it is common to use metric learning heads (e.g. ProtoNets (Snell et al., 2017)), which have no or few learnable parameters. Meta-Learning systems that employ a metric learning head are normally trained with an episodic training regime (Vinyals et al., 2016). Some of these approaches (e.g. TADAM (Oreshkin et al., 2018), FLUTE (Triantafillou et al., 2021), and Simple CNAPs (Bateni et al., 2020) use both a metric head and FiLM layers to adapt the backbone. FIT differs from all of the preceding approaches by using a powerful Naive Bayes metric head that uses episodic fine-tuning in the context of transfer learning, as opposed to the usual meta-learning. We show in Fig. 2a and Section 4 that the episodically fine-tuned Naive Bayes head consistently outperforms a conventional batch trained linear head in the low-shot transfer learning setting.

## 4 EXPERIMENTS

In this section, we evaluate the classification accuracy and updateable parameter efficiency of FIT in a series of challenging benchmarks and application scenarios. In all experiments, we use Big Transfer (BiT) (Kolesnikov et al., 2019), a leading, scalable, general purpose transfer learning algorithm as a point of comparison. First, we compare different variations of FIT to BiT on several standard downstream datasets in the few-shot regime. Second, we evaluate FIT against BiT on VTAB-1k (Zhai et al., 2019), which is arguably the most challenging transfer learning benchmark. Additionally, we compare FIT to the latest vision transformer based methods that have reported the highest accuracies on VTAB-1k to date. Third, we show how FIT can be used in a personalization scenario on the ORBIT (Massiceti et al., 2021) dataset, where a smaller updateable model is an important evaluation metric. Finally, we apply FIT to a few-shot federated learning scenario where minimizing the number of parameter updates and their size is a key requirement. Training and evaluation details are in Appendix A.11. Source code for experiments can be found at: `https://github.com/cambridge-mlg/fit`.

### 4.1 FEW-SHOT RESULTS

Fig. 3 shows the classification accuracy as a function of updateable parameters for FIT-LDA and BiT on four downstream datasets (CIFAR10, CIFAR100 (Krizhevsky et al., 2009), Pets (Parkhi et al., 2012), and Flowers (Nilsback & Zisserman, 2008)) that were used to evaluate the performance of BiT (Kolesnikov et al., 2019). Table A.1 contains complete tabular results with additional variants of FIT and BiT. All methods use the BiT-M-R50x1 (Kolesnikov et al., 2019) backbone that is pretrained on the ImageNet-21K (Russakovsky et al., 2015) dataset. The key observations from Fig. 3 are:

• For $\leq$10 shots (except 1-shot on CIFAR100), FIT-LDA outperforms BiT, often by a large margin.

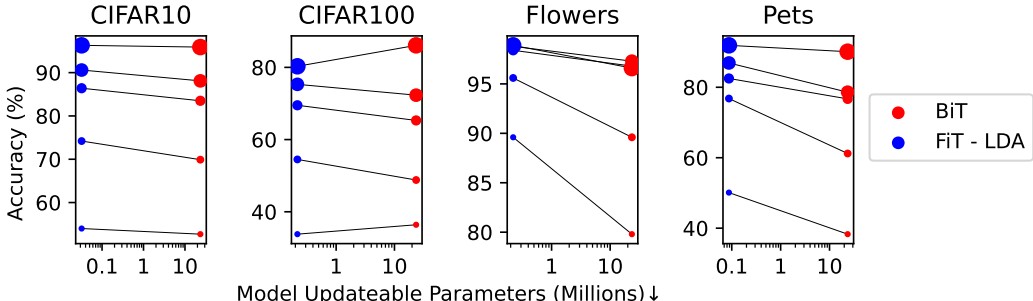

**Figure 3: FIT-LDA outperforms BiT at low-shot.** Classification accuracy as a function of the number of updateable parameters (log scale) and shots per class for FIT-LDA and BiT on four downstream datasets. Classification accuracy is on the vertical axis and is the average of 3 runs with different data sampling seeds. The dot size from smallest to largest indicates the number of shots per class — 1, 2, 5, 10, and All. A tabular version with results for additional variants is in Table A.1.

- On 3 out of 4 datasets, FIT-LDA outperforms BiT even when all of $\mathcal{D}$ is used for fine-tuning.
- FIT-LDA outperforms BiT despite BiT having more than 100 times as many updateable parameters.
- To avoid overfitting when $\mathcal{D}$ is small, Table A.2 indicates that it is better to split $\mathcal{D}$ into two disjoint partitions $\mathcal{D}_{train}$ and $\mathcal{D}_{test}$ and that $\mathcal{D}_S^\tau$ and $\mathcal{D}_Q^\tau$ should be randomly sub-sampled as opposed to using all of the data in each training iteration.

The datasets used in this section were similar in content to the dataset used for pretraining and the performance of FIT-QDA and FIT-LDA was similar, indicating that the covariance per class was not that useful for these datasets. In the next section, we test on a wider variety of datasets, many of which differ greatly from the upstream data.

## 4.2 VTAB-1K RESULTS

The VTAB-1k benchmark (Zhai et al., 2019) is a low to medium-shot transfer learning benchmark that consists of 19 datasets grouped into three distinct categories (natural, specialized, and structured). From each dataset, 1000 examples are drawn at random from the training split to use for the downstream dataset $\mathcal{D}$. After fine-tuning, the entire test split is used to evaluate classification performance. Table 2 shows the classification accuracy and updateable parameter count for the three variants of FIT and BiT (see Table A.3 for error bars). The key observations from our results are:

- Both FIT-QDA and FIT-LDA outperform BiT on VTAB-1k.
- The FIT-QDA variant has the best overall performance, showing that the class covariance is important to achieve superior results on datasets that differ from those used in upstream pretraining (e.g. the structured category of datasets). However, the updateable parameter cost is high.
- FIT-LDA utilizes two orders of magnitude fewer updateable parameters compared to BiT, making it the preferred approach.
- Table A.4 shows that it is best to use all of $\mathcal{D}$ for each of $\mathcal{D}_{train}$ and $\mathcal{D}_{test}$ (i.e. no split) and that $\mathcal{D}_S^\tau$ and $\mathcal{D}_Q^\tau$ should be episodically sub-sampled as opposed to using all of the data in each iteration.

**Table 2: FIT outperforms BiT on VTAB-1k.** Classification accuracy and updateable parameter count (with 10 classes) for FIT variants and BiT on VTAB-1k with BiT-M-R50x1 backbone. Accuracy figures are percentages. Bold type indicates the highest scores. Green indicates summary columns.

| | | | Natural | | | | | | | Specialized | | | | Structured | | | | | | | |
|---|---|---|---|---|---|---|---|---|---|---|---|---|---|---|---|---|---|---|---|---|---|
| Method | Params (M) ↓ | Overall Acc ↑ | Caltech101 | CIFAR100 | Flowers102 | Pets | Sun397 | SVHN | DTD | EuroSAT | Resisc45 | Camelyon | Retinopathy | Clevr-count | Clevr-dist | dSprites-loc | dSprites-ori | sNORB-azi | sNORB-elev | DMLab | KITTI-dist |
| BiT | 23.5 | 68.3 | 88.0 | 70.1 | 98.6 | 88.4 | 48.0 | 73.0 | **72.7** | 95.3 | **85.9** | 69.3 | **77.2** | 54.6 | 47.9 | **91.6** | **65.9** | 18.7 | 25.8 | **47.1** | **80.1** |
| FiT-QDA | 21.0 | **70.6** | 90.3 | 74.1 | **99.1** | **91.0** | 51.1 | **75.1** | 70.9 | **95.6** | 82.6 | 80.7 | 70.4 | 87.1 | 58.1 | 77.1 | 56.7 | **18.9** | **40.4** | 43.8 | 77.5 |
| FiT-LDA | 0.03 | 69.3 | **90.4** | **74.2** | 99.0 | 90.5 | **51.6** | 74.2 | 70.9 | 95.1 | 82.5 | **82.5** | 66.2 | 85.6 | 56.1 | 74.8 | 51.3 | 16.2 | 37.0 | 41.6 | 77.7 |
| FiT-ProtoNets | 0.03 | 65.5 | 89.6 | 73.9 | 98.6 | 90.8 | 51.5 | 50.1 | 68.2 | 93.8 | 77.0 | 79.9 | 57.9 | **88.7** | **58.3** | 68.6 | 34.2 | 13.5 | 35.0 | 39.3 | 75.3 |

Table 3 shows that FIT-LDA achieves state-of-the-art classification accuracy when compared to leading transfer learning methods pretrained on ImageNet-21k, while requiring the smallest number of updateable parameters and using the smallest backbone. All competing methods use a linear head.

Table 3: **FIT achieves SOTA on VTAB-1k**. Classification accuracy (%) for the 3 VTAB-1k categories (*Natural*, *Specialized*, and *Structured*) and mean accuracy over all 19 datasets (*Overall Acc*) and updateable parameter count (*Params*) for leading transfer learning methods using various backbones (backbone parameter count shown in parentheses) (Kolesnikov et al., 2019; Dosovitskiy et al., 2020; Tan & Le, 2021) pretrained on ImageNet-21k. ViT-Base-16 results from (Jie & Deng, 2022). BiT results from (Kolesnikov et al., 2020). Green indicates results summary columns.

| Method | Backbone | Params (M) ↓ | Overall Acc ↑ | Natural ↑ | Specialized ↑ | Structured ↑ |
|---|---|---|---|---|---|---|
| BiT (Kolesnikov et al., 2019) | BiT-M-R101x3 (382M) | 382 | 72.7 | 80.3 | **85.8** | 59.4 |
| BiT (Kolesnikov et al., 2019) | BiT-M-R152x4 (928M) | 928 | 73.5 | 80.8 | 85.7 | 61.1 |
| VPT (Jia et al., 2022) | ViT-Base-16 (85.8M) | 0.5 | 69.4 | 78.5 | 82.4 | 55.0 |
| Adapter (Houlsby et al., 2019) | ViT-Base-16 (85.8M) | 0.2 | 71.4 | 79.0 | 84.1 | 58.5 |
| AdaptFormer (Chen et al., 2022) | ViT-Base-16 (85.8M) | 0.2 | 72.3 | 80.6 | 84.9 | 58.8 |
| LoRA (Hu et al., 2021) | ViT-Base-16 (85.8M) | 0.3 | 72.3 | 79.5 | 84.6 | 59.8 |
| NOAH (Zhang et al., 2022) | ViT-Base-16 (85.8M) | 0.4 | 73.2 | 80.3 | 84.9 | 61.3 |
| Convpass (Jie & Deng, 2022) | ViT-Base-16 (85.8M) | 0.3 | 74.4 | 81.7 | 85.3 | 62.7 |
| FiT-LDA (ours) | EfficientNetV2-M (52.9M) | **0.15** | **74.9** | **82.2** | 84.3 | **63.7** |

## 4.3 PERSONALIZATION

In the personalization experiments, we use ORBIT (Massiceti et al., 2021), a real-world few-shot video dataset recorded by people who are blind/low-vision. A user collects a series of short videos of objects that they would like to recognize. The collected videos and associated labels are then uploaded to a central service to train a personalized classification model for that user. Once trained, the personalized model is downloaded to the user's smartphone. In this setting, models with a smaller number of updateable parameters are preferred in order to save model storage space on the central server and in transmitting any updated models to a user. The personalization is performed by taking a large pretrained model and adapting it using user's individual data. We follow the object recognition benchmark task proposed by the authors, which tests a personalized model on two different video types: *clean* where only a single object is present and *clutter* where that object appears within a realistic, multi-object scene.

In Table 4, we compare FIT-LDA to several competitive transfer learning and meta-learning methods that benchmark on the ORBIT dataset. We use the LDA variant of FIT, as it achieves higher accuracy in comparison to the ProtoNets variant, while using far fewer updateable parameters than QDA. As baselines, for transfer learning, we include FineTuner (Yosinski et al., 2014), which freezes the weights in the backbone and fine-tunes only the linear classifier head on an individual's data. For meta-learning approaches, we include ProtoNets (Snell et al., 2017) and Simple CNAPs (Bateni et al., 2020), which are meta-trained on Meta-Dataset (Dumoulin et al., 2021). In the lower part of Table 4, we compare FIT-LDA and BiT. Both models use the same BiT-M-R50x1 backbone pretrained on ImageNet-21K. Training and evaluation details are in Appendix A.11.2. For this comparison, we show frame and video accuracy, averaged over all the videos from all tasks across all test users (17 test users, 85 tasks in total). We also report the number of shared and individual updateable parameters required to be stored or transmitted. The key observations from our results are:

- FIT-LDA outperforms competitive meta-learning methods, Simple CNAPs and ProtoNets.
- FIT-LDA also outperforms FineTuner in terms of the video accuracy and performs within error bars of it in terms of the frame accuracy.
- The number of individual parameters for FIT-LDA is far fewer than in Simple CNAPs and BiT, and is of the same order of magnitude as FineTuner and ProtoNets.
- FIT-LDA pretrained on ImageNet-21K performs on par with BiT, while having orders of magnitude fewer updatable parameters.

## 4.4 FEW-SHOT FEDERATED LEARNING

We now show how FIT can be used in the few-shot federated learning setting, where training data are split between client nodes, e.g. mobile phones or personal laptops, and each client has only a handful of samples. Model training is performed via numerous communication rounds between a server and clients. In each round, the server selects a fraction of clients making updates and then sends the current model parameters to these clients. Clients update models locally using only their personal data and then send their parameter updates back to the server. Finally, the server aggregates information from all clients, updates the shared model parameters, and proceeds to the next round until

Table 4: **FIT outperforms competitive methods on ORBIT.** Average accuracy (95% confidence interval) over 85 test tasks. $b_\theta(x)$ is the backbone used. EN-1K(-MD) is EfficientNet-B0 pretrained on ImageNet (Meta-Dataset). RN-21K is BiT-M-R50x1 pretrained on ImageNet-21K. *Shared* is the number of parameters shared among all users. *Per User* is the number of parameters stored for each user with $C$ classes. *Average* is the mean number of individual user parameters over ORBIT.

| | | Clean Videos | | Clutter Videos | | Parameters | | |
|---|---|---|---|---|---|---|---|---|
| MODEL | $b_\theta(x)$ | FRAME ACC ↑ | VIDEO ACC ↑ | FRAME ACC ↑ | VIDEO ACC ↑ | SHARED ↓ | PER USER ↓ | AVERAGE ↓ |
| FineTuner | EN-1K | 78.1 (2.0) | 85.9 (2.3) | 63.1 (1.8) | 66.9 (2.4) | 4.01M | $(d_b+1)C$ | **0.01M** |
| Simple CNAPs | EN-MD | 73.1 (2.1) | 80.7 (2.6) | 61.6 (1.8) | 67.1 (2.4) | 5.67M | $\frac{d_b(d_b+3)}{2}C$ | 7.63M |
| ProtoNets | EN-MD | 71.6 (2.2) | 78.9 (2.7) | 63.0 (1.8) | 67.3 (2.4) | 4.01M | $d_b C$ | **0.01M** |
| FIT-LDA | EN-1K | 81.8 (1.8) | 90.6 (1.9) | 65.7 (1.8) | 70.7 (2.3) | 4.01M | $(d_b+1)C+|\psi|$ | **0.03M** |
| BiT | RN-21K | **82.4 (1.8)** | **89.7 (2.0)** | **67.7 (1.8)** | **73.9 (2.2)** | 0 | $(d_b+1)C+|\theta|$ | 23.5M |
| FIT-LDA | RN-21K | **83.5 (1.7)** | **90.1 (2.0)** | 67.4 (1.8) | 73.5 (2.2) | 23.5M | $(d_b+1)C+|\psi|$ | **0.03M** |

convergence. In this setting, models with a smaller number of updateable parameters are preferred in order to reduce the client-server communication cost which is typically bandwidth-limited. The configuration and training protocol of FIT help significantly reduce communication cost compared to standard fine-tuning protocols, as i) only FiLM layers are transferred at each communication round, ii) in contrast to standard linear head, the Naive Bayes head is not transferred and is constructed locally for each client. Table 5 shows communication parameters savings of FIT compared to BiT.

**Experiments** We use CIFAR100 (Krizhevsky et al., 2009), a relatively large-scale dataset compared to those commonly used to benchmark federated learning methods (Reddi et al., 2021; Shamsian et al., 2021). We employ the basic `FedAvg` (McMahan et al., 2016) algorithm, although other algorithms can also be used (Table A.7). We train all models for 60 communication rounds, with 5 clients per round and 10 update steps per client. Each client has 10 classes, which are sampled randomly before the start of training. We use a ProtoNets head for simplicity, although QDA and LDA heads could also be used. More specific training and evaluation details are in Appendix A.11.3.

We evaluate FIT in two scenarios, global and personalized. In the global setting, the aim is to construct a global classifier and report accuracy on the CIFAR100 test set. We assume that the server knows which classes belong to each client, and constructs a shared classifier by taking a mean over prototypes produced by clients for a particular class. In the personalized scenario, we test a personalized model on test classes present in the individual's training set and then report the mean accuracy over all clients. As opposed to the personalization experiments on ORBIT, where a personalized model is trained using only the client's local data, in this experiment we initialize a personalized model with the learned global FiLM parameters and then construct a ProtoNets classifier with an individual's data. Thus, the goal of the personalized setting is to estimate how advantageous distributed learning can be for training FiLM layers to build personalized few-shot models.

To test the global scenario, we compare FIT to BiT. As the training protocol used in BiT cannot be directly applied in the federated learning context, we trained BiT with a constant learning rate. We do not provide comparison on the personalized scenario, as BiT uses a linear classification head trained globally, as opposed to using only client's local data. As, to the best of our knowledge, there are no suitable federated learning systems to compare to, we define baselines which form an upper and lower bounds on the model performance. For the global scenario, we take a FIT model trained centrally on all available data as the upper bound baseline. To get the lower bound baseline, we train a FIT model for each client with their local data, then average FiLM parameters of these individual models and construct a global ProtoNets classifier using the resulting FiLM parameters. The upper bound is therefore standard batch training, the performance of which we hope federated learning can approach. The lower bound is a simplistic version of federated learning with a single communication round which federated averaging should improve over. For the personalized setting, the upper bound baseline is as in the global scenario from which we form a personalized classifier by taking a subset of classes belonging to a client from a global 100-way classifier. The lower bound baseline is set to a FIT model trained for each client individually. The upper bound is again standard batch training and the lower bound is derived from locally trained models which do not share information and therefore should be improved upon by federated learning.

**Results** Table 5 shows the comparison of FIT to BiT. Fig. 4 shows global and personalized classification accuracy as a function of communication cost for different numbers of clients and shots per

Table 5: **FıT is significantly more parameter efficient than BiT for federated learning.** Comparison of BiT and FıT in few-shot federated setting on CIFAR100 for different numbers of clients and shots per client. Global setting is used. Accuracy figures are percentages and the $\pm$ sign indicates the $95\%$ confidence interval over 3 runs. Parameter cost indicates the number of parameters sent in server-client communication. *Per round* is the number of parameters transmitted in each communication round and *Overall* is the number of parameters transmitted during the whole training.

| | Parameter cost | | 50 Clients | | | 100 Clients | | | 500 Clients | | |
|---|---|---|---|---|---|---|---|---|---|---|---|
| Models | Per round | Overall | 2-Shot | 5-Shot | 10-Shot | 2-Shot | 5-Shot | 10-Shot | 2-Shot | 5-Shot | 10-Shot |
| FıT | **0.1M** | **7M** | 68.5±0.3 | 73.8±0.7 | 77.1±0.3 | 74.4±0.3 | 76.5±0.2 | 78.4±0.1 | 77.9±0.4 | 77.6±0.1 | 79.2±0.3 |
| BiT | 237M | 14B | 69.0±1.4 | 75.6±0.6 | 78.9±0.6 | 73.2±1.9 | 78.6±0.3 | 80.2±0.3 | 69.9±0.3 | 77.3±0.9 | 78.6±1.3 |

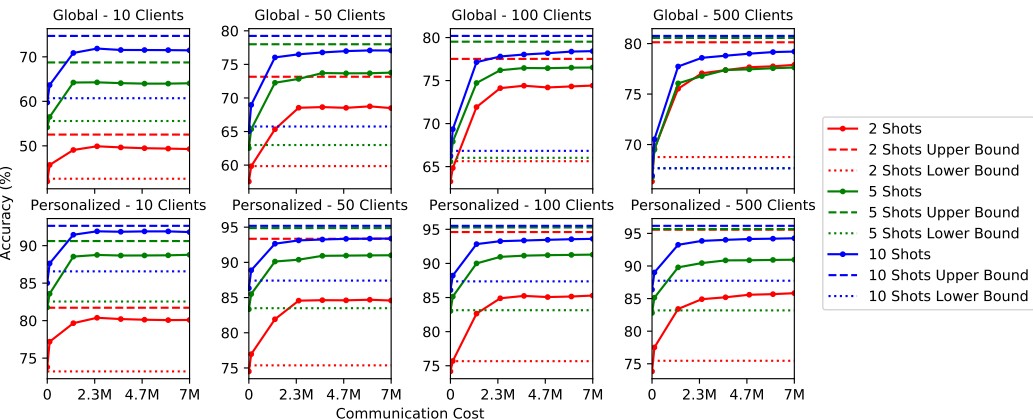

Figure 4: Global and personalized classification accuracy as a function of communication cost over 60 rounds for different numbers of clients and shots per client on CIFAR100. Classification accuracy is on the vertical axis and is the average of 3 runs with different data sampling seeds. The color of the line indicates the number of shots per class. The solid line shows the federated learning model, while dashed and dotted lines indicate the upper and lower bounds baselines, respectively.

client. By communication cost we mean the number of parameters transmitted during training. The key observations from our results are:

- FıT and BiT show comparable performance in terms of test accuracy, however FıT is much more parameter efficient and requires transmission of only $7M$ parameters through training, in contrast to $14B$ parameters for BiT. This makes FıT highly suitable for federated learning applications.

- In the global setting, the federated learning model is only slightly worse $(3-5\%)$ than the upper bound baseline, while outperforming the lower bound model, often by a large margin. This shows that FıT can be efficiently used in federated learning settings with different configurations.

- In the personalized scenario, for a sufficient number of clients $(\geq 50)$ the gap between the federated learning model and the upper bound model is significantly reduced with the increase in number of shots. Federated training strongly outperforms the lower bound baseline, surprisingly even in the case of 10 clients with disjoint classes. This provides empirical evidence that collaborative distributed training can be helpful for improving personalized models in the few-shot data regime.

In Appendix A.8, we show that distributed training of a FıT model can be efficiently used to learn from more extreme, non-natural image datasets like Quickdraw (Jongejan et al., 2016).

## 5 DISCUSSION

In this work, we proposed FıT, a parameter and data efficient few-shot transfer learning system that allows image classification models to be updated with only a small subset of the total model parameters. We demonstrated that FıT can outperform BiT using fewer than 1% of the updateable parameters and achieve state-of-the-art accuracy on VTAB-1k. We also showed the parameter efficiency benefits of employing FıT in model personalization and federated learning applications.

## ACKNOWLEDGMENTS

Aliaksandra Shysheya, John Bronskill, Massimiliano Patacchiola and Richard E. Turner are supported by an EPSRC Prosperity Partnership EP/T005386/1 between the EPSRC, Microsoft Research and the University of Cambridge. This work has been performed using resources provided by the Cambridge Tier-2 system operated by the University of Cambridge Research Computing Service `https://www.hpc.cam.ac.uk` funded by EPSRC Tier-2 capital grant EP/P020259/1. We thank the anonymous reviewers for key suggestions and insightful questions that significantly improved the quality of the paper. We also thank Aristeidis Panos and Siddharth Swaroop for providing helpful comments and suggestions.

**Reproducibility Statement**   Source code for experiments can be found at: `https://github.com/cambridge-mlg/fit`. The README file details the data preparation steps and includes the command lines to configure and run the experiments. Appendix A.11 details the training and evaluation procedures and hyperparameter settings for all of the experiments including few-shot and VTAB-1k transfer learning experiments, personalization on ORBIT experiments, and the federated learning experiments.

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

# A APPENDIX

## A.1 FILM LAYER PLACEMENT

Fig. A.1a illustrates a FiLM layer operating on a convolutional layer, and Fig. A.1b illustrates how a FiLM layer can be added to a ResNetV2 network block. FiLM layers can be similarly added to EfficientNet based backbones, amongst others.

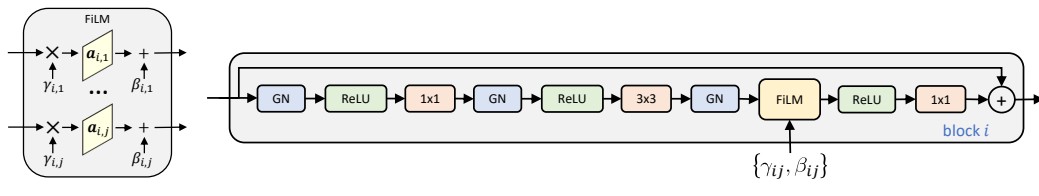

(a) A FiLM layer.          (b) A ResNet basic block with FiLM layers.

Figure A.1: (Left) A FiLM layer operating on convolutional feature maps in layer $i$ and channel $j$. (Right) How a FiLM layer is placed within a basic Residual network block (He et al., 2016a). GN is a Group Normalization layer, ReLU is a Rectified Linear Unit, and $1 \times 1$, and $3 \times 3$ are 2D convolutional layers with the stated kernel size.

## A.2 MODEL PARAMETERS

In this section, we provide details on the updateable parameter calculations for the LDA and QDA variants of BiT. Refer to Table 1.

For FIT-QDA and FIT-LDA, the means and covariances contribute to the updateable parameter count. We use a mean for every class which contributes $Cd_b$ updateable parameters. A covariance matrix has $d_b \times d_b$ values, however it can be represented in Cholesky factorized form which results in a lower triangular matrix and thus can be represented with $d_b(d_b + 1)/2$ values, with the rest being zeros.

In the case of FIT-LDA, where the covariance matrix is shared across all classes, a more compact representation is possible, resulting in considerable savings in updateable parameters:

$$
\begin{aligned}
p(y^* = c | b_{\boldsymbol{\theta},\boldsymbol{\psi}}(\boldsymbol{x}^*), \boldsymbol{\pi}, \boldsymbol{\mu}, \Sigma_{LDA}) &= \frac{\pi_c \mathcal{N}(b_{\boldsymbol{\theta},\boldsymbol{\psi}}(\boldsymbol{x}^*) | \mu_c, \Sigma_{LDA}))}{\sum_{c'}^C \pi_{c'} \mathcal{N}(b_{\boldsymbol{\theta},\boldsymbol{\psi}}(\boldsymbol{x}^*) | \mu_{c'}, \Sigma_{LDA})} \\
&= \frac{\pi_c \det(2\pi\Sigma_{LDA})^{-\frac{1}{2}} \exp\left(-\frac{1}{2}(b_{\boldsymbol{\theta},\boldsymbol{\psi}}(\boldsymbol{x}^*) - \mu_c)^T \Sigma_{LDA}^{-1}(b_{\boldsymbol{\theta},\boldsymbol{\psi}}(\boldsymbol{x}^*) - \mu_c)\right)}{\sum_{c'}^C \pi_{c'} \det(2\pi\Sigma_{LDA})^{-\frac{1}{2}} \exp\left(-\frac{1}{2}(b_{\boldsymbol{\theta},\boldsymbol{\psi}}(\boldsymbol{x}^*) - \mu_{c'})^T \Sigma_{LDA}^{-1}(b_{\boldsymbol{\theta},\boldsymbol{\psi}}(\boldsymbol{x}^*) - \mu_{c'})\right)} \\
&= \frac{\pi_c \exp\left(-\frac{1}{2}b_{\boldsymbol{\theta},\boldsymbol{\psi}}(\boldsymbol{x}^*)^T \Sigma_{LDA}^{-1} b_{\boldsymbol{\theta},\boldsymbol{\psi}}(\boldsymbol{x}^*) + \mu_c^T \Sigma_{LDA}^{-1} b_{\boldsymbol{\theta},\boldsymbol{\psi}}(\boldsymbol{x}^*) - \frac{1}{2}\mu_c^T \Sigma_{LDA}^{-1} \mu_c\right)}{\sum_{c'}^C \pi_{c'} \exp\left(-\frac{1}{2}b_{\boldsymbol{\theta},\boldsymbol{\psi}}(\boldsymbol{x}^*)^T \Sigma_{LDA}^{-1} b_{\boldsymbol{\theta},\boldsymbol{\psi}}(\boldsymbol{x}^*) + \mu_{c'}^T \Sigma_{LDA}^{-1} b_{\boldsymbol{\theta},\boldsymbol{\psi}}(\boldsymbol{x}^*) - \frac{1}{2}\mu_{c'}^T \Sigma_{LDA}^{-1} \mu_{c'}\right)} \\
&= \frac{\pi_c \exp\left(\mu_c^T \Sigma_{LDA}^{-1} b_{\boldsymbol{\theta},\boldsymbol{\psi}}(\boldsymbol{x}^*) - \frac{1}{2}\mu_c^T \Sigma_{LDA}^{-1} \mu_c\right)}{\sum_{c'}^C \pi_{c'} \exp\left(\mu_{c'}^T \Sigma_{LDA}^{-1} b_{\boldsymbol{\theta},\boldsymbol{\psi}}(\boldsymbol{x}^*) - \frac{1}{2}\mu_{c'}^T \Sigma_{LDA}^{-1} \mu_{c'}\right)}
\end{aligned}
\tag{A.1}
$$

From Eq. (A.1), it follows that to compute the probability of classifying a test point $\boldsymbol{x}^*$, we need to store $\mu_c^T \Sigma_{LDA}^{-1}$ which has dimensionality $d_b$ and $\mu_c^T \Sigma_{LDA}^{-1} \mu_c$ which has dimensionality 1 for each class $c$, resulting in only $C(d_b + 1)$ parameters required for the FIT-LDA head. Since the covariance matrix is not shared in the case of FIT-QDA, no additional savings are possible in that case.

## A.3 VTAB-1K DATASETS

The VTAB-1k benchmark (Zhai et al., 2019) is a low to medium-shot transfer learning benchmark that consists of 19 datasets grouped into three distinct categories (natural, specialized, and structured).

The natural datasets are: Caltech101 (Fei-Fei et al., 2006), CIFAR100 (Krizhevsky et al., 2009), Flowers102 (Nilsback & Zisserman, 2008), Pets (Parkhi et al., 2012), Sun397 (Xiao et al., 2010), SVHN (Netzer et al., 2011), and DTD (Cimpoi et al., 2014).

The specialized datasets are: EuroSAT (Helber et al., 2019), Resics45 (Cheng et al., 2017),Patch Camelyon (Veeling et al., 2018), and Retinopathy (Kaggle & EyePacs, 2015).

The structured datasets are: CLEVR-count (Johnson et al., 2017), CLEVR-dist (Johnson et al., 2017), dSprites-loc (Matthey et al., 2017), dSprites-ori (Matthey et al., 2017), SmallNORB-azi (LeCun et al., 2004), SmallNORB-elev (LeCun et al., 2004), DMLab (Beattie et al., 2016), KITTI-dist (Geiger et al., 2013).

## A.4 ADDITIONAL RESULTS

This section contains additional results that would not fit into the main paper, including tabular versions of figures. Note, in some results, we use a new metric *Relative Model Update Size* or RMUS, which is the ratio of the number of updateable parameters in one model to another. In our experiments, we measure RMUS relative to the number of updatable parameters in BiT model.

### A.4.1 ADDITIONAL FEW-SHOT RESULTS

Table A.1 shows the tabular version of Fig. 3. In addition, Table A.1 includes results for an additional variant of BiT (BiT-FiLM), and two additional variants of FIT (FIT-QDA and FIT-ProtoNets). BiT-FiLM is a variant of BiT that uses the same training protocol as the standard version of BiT, but the backbone weights $\theta$ are frozen and FiLM layers are added in the same manner as FIT. The FiLM parameters $\psi$ and the linear head weights $\phi$ are learned during training. The results are shown in Table A.1 and the key observations are:

- In general, at low-shot, the standard version of BiT outperforms BiT-FiLM. However, as the shot increases, especially when training on all of $\mathcal{D}$, BiT-FiLM is equal in classification accuracy.
- The above implies that FiLM layers have sufficient capacity to accurately fine-tune to downstream datasets, but the FIT head and training protocol are needed to achieve superior results.
- While the accuracy of FIT-QDA and FIT-LDA is similar, the storage requirements for a covariance matrix for each class makes QDA impractical if model update size is an important consideration.
- The accuracy of FIT-ProtoNets is slightly lower than FIT-LDA, but often betters BiT, despite BiT having more than 100 times as many updateable parameters.

Table A.1: Classification accuracy and Relative Model Update Size (RMUS) for all three variants of FIT and two variants of BiT on standard downstream datasets as a function of shots per class. The backbone is BiT-M-R50x1 with $|\theta| = 23,500,352$, $\psi = 11,648$, and $d_b = 2048$. Accuracy figures are percentages and the $\pm$ sign indicates the 95% confidence interval over 3 runs. Bold type indicates the highest scores (within the confidence interval).

| | | | BiT | | | | FiLM Transfer (FiT) | | | | | |
| | | | Standard | | FiLM | | QDA | | LDA | | ProtoNets | |
| Dataset | $C$ | Shot | Accuracy | RMUS | Accuracy | RMUS | Accuracy | RMUS | Accuracy | RMUS | Accuracy | RMUS |
|---|---|---|---|---|---|---|---|---|---|---|---|---|
| | 10 | 1 | **52.7±3.9** | 1.0 | 47.0±14.2 | 0.0014 | 54.0±5.7 | 0.893 | **54.0±5.7** | 0.0014 | 52.9±5.0 | 0.0014 |
| | 10 | 2 | **69.9±2.6** | 1.0 | 63.4±3.6 | 0.0014 | 73.0±8.8 | 0.893 | 74.2±8.8 | 0.0014 | 68.9±9.4 | 0.0014 |
| CIFAR10 | 10 | 5 | **83.5±3.0** | 1.0 | 82.8±1.9 | 0.0014 | 85.4±3.9 | 0.893 | 86.4±4.2 | 0.0014 | 81.8±5.1 | 0.0014 |
| (Krizhevsky et al., 2009) | 10 | 10 | **88.1±2.0** | 1.0 | 46.1±39.7 | 0.0014 | 89.5±1.3 | 0.893 | 90.6±1.0 | 0.0014 | 87.3±2.3 | 0.0014 |
| | 10 | All | 95.9±0.2 | 1.0 | 96.0±0.2 | 0.0014 | **96.4±0.0** | 0.893 | 96.3±0.1 | 0.0014 | 96.0±0.1 | 0.0014 |
| | 100 | 1 | **36.4±1.3** | 1.0 | 30.2±2.7 | 0.0091 | 33.8±0.8 | 8.860 | 33.8±0.8 | 0.0091 | 30.7±0.7 | 0.0091 |
| | 100 | 2 | 48.8±0.2 | 1.0 | 43.0±2.9 | 0.0091 | **55.1±1.1** | 8.860 | 54.5±1.0 | 0.0091 | 50.6±1.8 | 0.0091 |
| CIFAR100 | 100 | 5 | 65.3±1.5 | 1.0 | 39.6±0.6 | 0.0091 | 69.0±0.7 | 8.860 | 69.5±1.3 | 0.0091 | 67.9±0.7 | 0.0091 |
| (Krizhevsky et al., 2009) | 100 | 10 | 72.3±0.9 | 1.0 | 50.1±0.2 | 0.0091 | 75.6±0.6 | 8.860 | 75.3±0.6 | 0.0091 | 74.7±0.3 | 0.0091 |
| | 100 | All | **86.1±0.1** | 1.0 | 82.6±0.2 | 0.0091 | 82.1±0.1 | 8.860 | 82.6±0.2 | 0.0091 | 80.3±0.4 | 0.0091 |
| | 102 | 1 | 79.8±0.7 | 1.0 | 79.3±2.5 | 0.0093 | **89.6±0.9** | 9.036 | **89.6±0.9** | 0.0093 | 85.1±0.9 | 0.0093 |
| | 102 | 2 | 89.6±1.4 | 1.0 | 91.7±0.8 | 0.0093 | **95.6±0.5** | 9.036 | **95.6±0.7** | 0.0093 | 93.0±0.9 | 0.0093 |
| Flowers102 | 102 | 5 | 96.8±0.7 | 1.0 | 96.3±0.2 | 0.0093 | **98.3±0.3** | 9.036 | **98.4±0.3** | 0.0093 | **98.0±0.3** | 0.0093 |
| (Nilsback & Zisserman, 2008) | 102 | 10 | 97.3±0.7 | 1.0 | 96.9±0.7 | 0.0093 | **99.0±0.1** | 9.036 | 98.8±0.1 | 0.0093 | 98.6±0.1 | 0.0093 |
| | 102 | All | 96.6±0.4 | 1.0 | 96.9±0.4 | 0.0093 | **99.0±0.1** | 9.036 | **98.9±0.1** | 0.0093 | 98.6±0.2 | 0.0093 |
| | 37 | 1 | 38.3±2.3 | 1.0 | 36.8±5.5 | 0.0037 | 50.2±2.8 | 3.297 | 50.1±3.1 | 0.0037 | 46.6±2.0 | 0.0037 |
| | 37 | 2 | 61.2±2.6 | 1.0 | 60.1±3.0 | 0.0037 | **74.8±1.4** | 3.297 | **76.8±1.8** | 0.0037 | **72.8±2.7** | 0.0037 |
| Pets | 37 | 5 | 76.7±2.0 | 1.0 | 76.8±1.5 | 0.0037 | 80.0±0.2 | 3.297 | **82.5±0.4** | 0.0037 | 78.4±0.7 | 0.0037 |
| (Parkhi et al., 2012) | 37 | 10 | 78.6±5.3 | 1.0 | 79.2±4.4 | 0.0037 | 86.2±1.0 | 3.297 | 86.9±1.5 | 0.0037 | 85.8±0.4 | 0.0037 |
| | 37 | All | 90.1±0.6 | 1.0 | 90.1±0.2 | 0.0037 | 91.7±0.3 | 3.297 | **91.9±0.3** | 0.0037 | 91.2±0.1 | 0.0037 |

### A.4.2 FEW-SHOT TASK ABLATIONS

Table A.2 shows the few-shot results for all three variants of FIT with different ablations on how the downstream dataset $\mathcal{D}$ is allocated during training. *No Split* indicates that $\mathcal{D}_{train}$ is not split into two disjoint partitions and $\mathcal{D}_{train} = \mathcal{D}_{test} = \mathcal{D}$. However, $\mathcal{D}_{train}$ and $\mathcal{D}_{test}$ are sampled to form episodic fine-tuning tasks as detailed in Algorithm A.2. *Split* indicates that $\mathcal{D}_{train}$ is split into two disjoint partitions as detailed in Algorithm A.1 and then sampled into tasks as described in Algorithm A.2. *Use All* indicates that $\mathcal{D}_{train} = \mathcal{D}_{test} = \mathcal{D}$ (i.e. $\mathcal{D}$ is not split) and that $\mathcal{D}_{train}$ and $\mathcal{D}_{test}$ are not sampled and that $\mathcal{D}_{\mathcal{S}}^{\tau} = \mathcal{D}_{\mathcal{Q}}^{\tau} = \mathcal{D}$ for all tasks $\tau$.

Table A.2 shows that *Use All* is consistently the worst option. In general, in the few-shot case, *Split* either outperforms *No Split* (CIFAR10, Pets) or achieves the same level of performance (CIFAR100, Flowers102). As a result, we use the *Split* option when reporting the few-shot results.

Table A.2: Classification accuracy for all three variants of FIT as a function of shots per class and how the downstream dataset $\mathcal{D}$ is utilized during training on standard datasets. The backbone is BiT-M-R50x1 with $|\boldsymbol{\theta}| = 23,500,352$, $\boldsymbol{\psi} = 11,648$, and $d_b = 2048$. Accuracy figures are percentages and the $\pm$ sign indicates the 95% confidence interval over 3 runs.

| Dataset | Shot | QDA | | | LDA | | | ProtoNets | | |
|---|---|---|---|---|---|---|---|---|---|---|
| | | No Split | Split | Use All | No Split | Split | Use All | No Split | Split | Use All |
| CIFAR10 (Krizhevsky et al., 2009) | 1 | 42.5±2.2 | 54.0±5.7 | 37.2±3.9 | 36.7±4.3 | 54.0±5.7 | 28.5±2.3 | 53.0±5.1 | 52.9±5.0 | 53.0±5.1 |
| | 2 | 62.8±4.8 | 73.0±8.8 | 68.2±5.4 | 60.4±1.6 | 74.2±8.8 | 40.8±5.2 | 65.2±4.5 | 68.9±9.4 | 65.2±4.5 |
| | 5 | 79.7±0.9 | 85.4±3.9 | 79.6±1.0 | 86.6±3.5 | 86.4±4.2 | 69.4±3.5 | 76.1±4.2 | 81.8±5.1 | 75.6±9.3 |
| | 10 | 84.2±0.1 | 89.5±1.3 | 84.0±0.3 | 92.3±0.3 | 90.6±1.0 | 87.3±0.9 | 84.5±3.8 | 87.3±2.3 | 81.9±4.5 |
| | All | 96.6±0.1 | 96.4±0.0 | 96.2±0.0 | 96.6±0.0 | 96.3±0.1 | 96.6±0.0 | 96.1±0.1 | 96.0±0.1 | 95.4±0.0 |
| CIFAR100 (Krizhevsky et al., 2009) | 1 | 33.0±3.9 | 33.8±0.8 | 33.0±0.8 | 32.8±2.7 | 33.8±0.8 | 33.8±0.8 | 30.7±0.7 | 30.7±0.7 | 30.7±0.7 |
| | 2 | 54.5±2.0 | 55.1±1.1 | 45.6±2.0 | 55.6±1.2 | 54.5±1.0 | 45.1±3.1 | 52.3±2.0 | 50.6±1.8 | 40.2±6.1 |
| | 5 | 69.7±1.0 | 69.0±0.7 | 57.9±0.4 | 69.8±1.0 | 69.5±1.3 | 57.6±1.8 | 68.7±1.5 | 67.9±0.7 | 56.6±4.2 |
| | 10 | 75.5±0.3 | 75.6±0.6 | 67.6±7.9 | 75.6±0.2 | 75.3±0.6 | 67.0±3.2 | 74.8±0.2 | 74.7±0.3 | 65.4±2.8 |
| | All | 82.4±0.1 | 82.1±0.1 | 77.2±0.1 | 82.6±0.2 | 82.6±0.2 | 81.1±0.1 | 80.6±0.2 | 80.3±0.4 | 78.2±0.1 |
| Flowers102 (Nilsback & Zisserman, 2008) | 1 | 86.1±0.5 | 89.6±0.9 | 89.1±0.8 | 82.1±0.8 | 89.6±0.9 | 89.6±0.9 | 85.1±0.9 | 85.1±0.9 | 85.1±0.9 |
| | 2 | 95.2±0.6 | 95.6±0.5 | 94.4±0.6 | 95.6±0.5 | 95.6±0.7 | 94.9±0.5 | 93.9±1.0 | 93.0±0.9 | 91.9±1.2 |
| | 5 | 98.4±0.2 | 98.3±0.3 | 98.2±0.4 | 98.5±0.2 | 98.4±0.3 | 97.4±0.4 | 98.1±0.4 | 98.0±0.3 | 96.6±0.6 |
| | 10 | 99.0±0.0 | 99.0±0.1 | 98.9±0.1 | 98.9±0.1 | 98.8±0.1 | 98.5±0.0 | 98.6±0.0 | 98.6±0.1 | 96.7±0.0 |
| | All | 99.1±0.1 | 99.0±0.1 | 98.8±0.0 | 99.0±0.1 | 98.9±0.1 | 98.4±0.0 | 98.8±0.1 | 98.6±0.2 | 96.7±0.0 |
| Pets (Parkhi et al., 2012) | 1 | 29.4±3.1 | 50.2±2.8 | 50.2±2.8 | 17.2±6.3 | 50.1±3.1 | 50.0±3.2 | 46.6±2.0 | 46.6±2.0 | 46.6±2.0 |
| | 2 | 53.0±2.1 | 74.8±1.4 | 64.2±0.7 | 49.4±5.7 | 76.8±1.8 | 53.4±2.0 | 60.1±0.4 | 72.8±2.7 | 60.1±0.4 |
| | 5 | 81.2±2.3 | 80.0±0.2 | 73.6±1.1 | 82.1±2.2 | 82.5±0.4 | 71.0±3.1 | 83.0±2.4 | 78.4±0.7 | 67.4±6.2 |
| | 10 | 87.3±1.0 | 86.2±1.0 | 80.0±6.2 | 87.1±0.8 | 86.9±1.5 | 81.6±1.6 | 87.1±1.2 | 85.8±0.4 | 79.2±2.0 |
| | All | 92.1±0.2 | 91.7±0.3 | 77.8±0.0 | 91.8±0.2 | 91.9±0.3 | 88.3±0.1 | 91.8±0.2 | 91.2±0.1 | 82.4±0.4 |

## A.5 ADDITIONAL VTAB-1K RESULTS

Table A.3 depicts a more comprehensive version of Table 2 including error bars and RMUS for each dataset.

### A.5.1 VTAB-1K TASK ABLATIONS

Table A.4 shows the VTAB-1k results for all three variants of FIT with different ablations on how the downstream dataset $\mathcal{D}$ is allocated during training. Refer to Appendix A.4.2 for the meanings of *No Split*, *Split*, and *Use All*. With some minor exceptions, the *Use All* case performs the worst. The performance of the *No Split* and *Split* options is very close, with *No Split* being slightly better when averaged over all of the datasets. As a result, we use the *No Split* option when reporting the VTAB-1k results.

Table A.5 compares the classification accuracy of the LDA and linear heads as a function of the learnable parameters in the backbone. When all parameters in the backbone are learnable and when FiLM layers are employed, the Naive Bayes LDA head outperforms the linear head. In the case when all of the parameters in the backbone are frozen, the linear head is superior since in that case the LDA head has only 2 learnable parameters (the covariance weights $\boldsymbol{e}$).

Table A.3: **FIT outperforms BiT on VTAB-1k.** Classification accuracy and Relative Model Update Size (RMUS) for all three variants of FIT and BiT on the VTAB-1k benchmark. The backbone is BiT-M-R50x1. Accuracy figures are percentages and the $\pm$ sign indicates the 95% confidence interval over 3 runs. Bold type indicates the highest scores (within the confidence interval).

| | | BiT | | FiLM Transfer (FiT) | | | | | | EffNetV2-M |
| | | | | QDA | | LDA | | ProtoNets | | |
| Dataset | $C$ | Accuracy↑ | RMUS↓ | Accuracy↑ | RMUS↓ | Accuracy↑ | RMUS↓ | Accuracy↑ | RMUS↓ | Accuracy↑ |
|---|---|---|---|---|---|---|---|---|---|---|
| Caltech101 | 102 | 88.0±0.2 | 1.0 | **90.3±0.8** | 9.04 | **90.4±0.8** | 0.0093 | 89.6±0.2 | 0.0093 | 96.1±0.1 |
| CIFAR100 | 100 | 70.1±0.1 | 1.0 | **74.1±0.1** | 8.86 | **74.2±0.5** | 0.0091 | 73.9±0.3 | 0.0091 | 81.3±0.2 |
| Flowers102 | 102 | 98.6±0.0 | 1.0 | **99.1±0.1** | 9.04 | **99.0±0.1** | 0.0093 | 98.6±0.0 | 0.0093 | 99.6±0.0 |
| Pets | 37 | 88.4±0.2 | 1.0 | **91.0±0.3** | 3.30 | 90.5±0.0 | 0.0037 | **90.8±0.2** | 0.0037 | 93.0±0.1 |
| Sun397 | 397 | 48.0±0.1 | 1.0 | **51.1±0.7** | 34.29 | **51.6±0.5** | 0.0339 | **51.5±1.4** | 0.0339 | 47.9±0.5 |
| SVHN | 10 | 73.0±0.2 | 1.0 | **75.1±1.3** | 0.89 | **74.2±0.9** | 0.0014 | 50.1±2.2 | 0.0014 | 85.1±1.1 |
| DTD | 47 | **72.7±0.3** | 1.0 | 70.9±0.1 | 4.18 | 70.9±0.1 | 0.0046 | 68.2±1.1 | 0.0046 | 72.4±0.6 |
| EuroSAT | 10 | 95.3±0.1 | 1.0 | **95.6±0.1** | 0.89 | 95.1±0.1 | 0.0014 | 93.8±0.1 | 0.0014 | 96.0±0.1 |
| Resics45 | 45 | **85.9±0.0** | 1.0 | 82.6±0.1 | 4.01 | 82.5±0.2 | 0.0044 | 77.0±0.0 | 0.0044 | 86.1±0.4 |
| Patch Camelyon | 2 | 69.3±0.8 | 1.0 | **80.7±1.2** | 0.18 | **82.5±0.7** | 0.0007 | 79.9±0.2 | 0.0007 | 82.9±1.1 |
| Retinopathy | 5 | **77.2±0.6** | 1.0 | 70.4±0.1 | 0.45 | 66.2±0.5 | 0.0009 | 57.9±0.3 | 0.0009 | 72.1±0.6 |
| CLEVR-count | 8 | 54.6±7.1 | 1.0 | 87.1±0.3 | 0.71 | 85.6±0.9 | 0.0012 | **88.7±0.3** | 0.0012 | 94.9±0.5 |
| CLEVR-dist | 6 | 47.9±0.8 | 1.0 | **58.1±0.8** | 0.54 | 56.1±0.8 | 0.0010 | **58.3±0.6** | 0.0010 | 60.9±2.0 |
| dSprites-loc | 16 | **91.6±1.1** | 1.0 | 77.1±2.0 | 1.43 | 74.8±1.4 | 0.0019 | 68.6±2.4 | 0.0019 | 94.8±0.1 |
| dSprites-ori | 16 | **65.9±0.3** | 1.0 | 56.7±0.3 | 1.43 | 51.3±0.7 | 0.0019 | 34.2±0.8 | 0.0019 | 60.8±1.1 |
| SmallNORB-azi | 18 | **18.7±0.2** | 1.0 | **18.9±0.6** | 1.61 | 16.2±0.1 | 0.0021 | 13.5±0.1 | 0.0021 | 16.9±1.0 |
| SmallNORB-elev | 9 | 25.8±0.9 | 1.0 | **40.4±0.2** | 0.80 | 37.0±0.6 | 0.0013 | 35.0±0.6 | 0.0013 | 53.5±1.3 |
| DMLab | 6 | **47.1±0.1** | 1.0 | 43.8±0.3 | 0.54 | 41.6±0.6 | 0.0010 | 39.3±0.3 | 0.0010 | 45.6±1.3 |
| KITTI-dist | 4 | **80.1±0.9** | 1.0 | 77.5±0.7 | 0.36 | 77.7±0.8 | 0.0008 | 75.3±0.2 | 0.0008 | 82.2±0.4 |
| All | | 68.3 | | **70.6** | | 69.3 | | 65.5 | | 74.9 |
| Natural | | 77.0 | | **78.8** | | 78.7 | | 74.7 | | 82.2 |
| Specialized | | 81.9 | | **82.3** | | 81.5 | | 77.1 | | 84.3 |
| Structured | | 54.0 | | **57.5** | | 55.0 | | 51.6 | | 63.7 |

Table A.4: Classification accuracy for all three variants of FIT on the VTAB-1k benchmark as a function of how the downstream dataset $\mathcal{D}$ is utilized during training. The backbone is BiT-M-R50x1. Accuracy figures are percentages and the $\pm$ sign indicates the 95% confidence interval over 3 runs. Bold type indicates the highest scores.

| | QDA | | | LDA | | | ProtoNets | | |
| Dataset | No Split | Split | Use All | No Split | Split | Use All | No Split | Split | Use All |
|---|---|---|---|---|---|---|---|---|---|
| Caltech101 (Fei-Fei et al., 2006) | 90.3±0.8 | 90.0±0.7 | 85.5±0.0 | 90.4±0.8 | 90.7±0.7 | 87.3±0.0 | 89.6±0.2 | 89.7±0.3 | 82.3±0.0 |
| CIFAR100 (Krizhevsky et al., 2009) | 74.1±0.1 | 74.8±0.3 | 63.4±0.0 | 74.2±0.5 | 74.1±0.4 | 69.0±0.0 | 73.9±0.3 | 73.7±0.6 | 65.5±0.2 |
| Flowers102 (Nilsback & Zisserman, 2008) | 99.1±0.1 | 99.0±0.1 | 98.8±0.0 | 99.0±0.1 | 98.9±0.1 | 98.5±0.0 | 98.6±0.0 | 98.6±0.1 | 96.7±0.0 |
| Pets (Parkhi et al., 2012) | 91.0±0.3 | 90.4±0.4 | 75.6±0.0 | 90.5±0.5 | 90.5±0.5 | 87.2±0.1 | 90.8±0.2 | 90.4±0.2 | 85.6±0.0 |
| Sun397 (Xiao et al., 2010) | 51.1±0.7 | 52.1±0.1 | 49.3±0.7 | 51.6±0.5 | 50.8±0.7 | 42.7±3.4 | 51.5±1.4 | 50.4±0.8 | 42.4±4.2 |
| SVHN (Netzer et al., 2011) | 75.1±1.3 | 73.3±1.4 | 26.2±0.1 | 74.2±0.9 | 71.5±0.2 | 67.4±0.0 | 50.1±2.2 | 47.4±1.4 | 35.1±0.0 |
| DTD (Cimpoi et al., 2014) | 70.9±0.1 | 70.2±0.3 | 72.8±0.0 | 70.9±0.1 | 70.8±0.3 | 66.5±0.0 | 68.2±1.1 | 68.4±1.0 | 61.3±0.2 |
| EuroSAT (Helber et al., 2019) | 95.6±0.1 | 94.7±0.6 | 93.5±0.0 | 95.1±0.1 | 94.3±0.6 | 94.3±0.0 | 93.8±0.1 | 92.7±0.2 | 89.4±0.1 |
| Resics45 (Cheng et al., 2017) | 82.6±0.1 | 82.0±0.3 | 77.5±0.0 | 82.5±0.2 | 80.8±0.2 | 78.3±0.1 | 77.0±0.0 | 76.4±0.8 | 71.9±0.1 |
| Patch Camelyon (Veeling et al., 2018) | 80.7±1.2 | 81.5±1.0 | 65.7±0.0 | 82.5±0.7 | 80.5±0.8 | 82.4±0.5 | 79.9±0.2 | 78.5±2.1 | 69.0±0.1 |
| Retinopathy (Kaggle & EyePacs, 2015) | 70.4±0.1 | 67.5±1.0 | 25.5±0.0 | 66.2±0.5 | 63.4±0.3 | 25.0±0.1 | 57.9±0.3 | 58.4±0.6 | 17.0±0.0 |
| CLEVR-count (Johnson et al., 2017) | 87.1±0.3 | 84.9±1.1 | 40.6±0.1 | 85.6±0.9 | 84.3±0.5 | 82.0±0.1 | 88.7±0.3 | 85.4±0.7 | 87.4±0.1 |
| CLEVR-dist (Johnson et al., 2017) | 58.1±0.8 | 58.4±0.6 | 39.1±0.1 | 56.1±0.8 | 55.7±1.7 | 57.7±0.4 | 58.3±0.6 | 55.0±1.2 | 33.7±0.2 |
| dSprites-loc (Matthey et al., 2017) | 77.1±2.0 | 75.1±1.2 | 13.5±0.3 | 74.8±1.4 | 71.1±1.1 | 62.4±1.1 | 68.6±2.4 | 66.8±0.8 | 74.8±1.4 |
| dSprites-ori (Matthey et al., 2017) | 56.7±0.3 | 55.6±0.7 | 39.6±1.8 | 51.3±0.7 | 48.0±0.1 | 53.8±0.0 | 34.2±0.8 | 32.4±1.2 | 36.7±0.3 |
| SmallNORB-azi (LeCun et al., 2004) | 18.9±0.6 | 19.8±0.6 | 14.0±0.1 | 16.2±0.1 | 17.1±1.3 | 15.8±0.3 | 13.5±0.1 | 13.0±0.6 | 13.1±0.0 |
| SmallNORB-elev (LeCun et al., 2004) | 40.4±0.2 | 40.3±1.4 | 28.2±0.1 | 37.0±0.6 | 38.5±1.6 | 36.1±0.3 | 35.0±0.6 | 34.0±0.9 | 26.5±0.1 |
| DMLab (Beattie et al., 2016) | 43.8±0.3 | 41.2±1.3 | 33.7±0.1 | 41.6±0.6 | 39.5±1.3 | 38.9±0.4 | 39.3±0.3 | 38.6±0.2 | 28.2±0.1 |
| KITTI-dist (Geiger et al., 2013) | 77.5±0.7 | 77.2±2.2 | 73.6±0.1 | 77.7±0.8 | 77.5±1.3 | 73.1±0.1 | 75.3±0.2 | 74.3±2.9 | 69.0±0.9 |
| All | **70.6** | 69.9 | 53.5 | **69.3** | 68.3 | 64.1 | **65.5** | 64.4 | 57.1 |
| Natural | **78.8** | 78.5 | 67.4 | **78.7** | 78.2 | 74.1 | **74.7** | 74.1 | 67.0 |
| Specialized | **82.3** | 81.4 | 65.6 | **81.5** | 79.7 | 70.0 | **77.1** | 76.5 | 61.8 |
| Structured | **57.5** | 56.6 | 35.3 | **55.0** | 54.0 | 52.5 | **51.6** | 49.9 | 46.2 |

Table A.5: Comparison of the classification accuracy of LDA and linear heads on VTAB-1k as a function of the learnable parameters in the backbone. "None" indicates that all of the parameters in the pre-trained backbone are frozen (i.e. no learnable parameters). "FiLM" indicates the that the only learnable parameters in the backbone are the FiLM layers. "All" indicates that all the parameters in backbone are learnable and that FiLM is not used. The backbone is BiT-M-R50x1. Accuracy figures are percentages and the $\pm$ sign indicates the 95% confidence interval over 3 runs. This is the full version of Fig. 2a

| Dataset | LDA | | | Linear | | |
|---|---|---|---|---|---|---|
| | None | FiLM | All | None | FiLM | All |
| Caltech101 (Fei-Fei et al., 2006) | 86.0±0.0 | 90.4±0.8 | 90.4±0.5 | 87.7±0.2 | 89.3±0.2 | 88.0±0.2 |
| CIFAR100 (Krizhevsky et al., 2009) | 64.8±0.0 | 74.2±0.5 | 63.3±0.9 | 61.4±0.1 | 70.2±0.1 | 70.1±0.1 |
| Flowers102 (Nilsback & Zisserman, 2008) | 98.7±0.0 | 99.0±0.1 | 98.9±0.2 | 98.6±0.0 | 98.7±0.1 | 98.6±0.0 |
| Pets (Parkhi et al., 2012) | 85.7±0.1 | 90.5±0.0 | 90.7±0.4 | 84.7±0.2 | 87.4±0.0 | 88.4±0.2 |
| Sun397 (Xiao et al., 2010) | 47.7±0.0 | 51.6±0.5 | 38.6±1.4 | 46.8±0.0 | 46.8±0.2 | 48.0±0.1 |
| SVHN (Netzer et al., 2011) | 40.9±0.1 | 74.2±0.9 | 85.1±2.3 | 42.8±1.2 | 54.9±0.4 | 73.0±0.2 |
| DTD (Cimpoi et al., 2014) | 72.0±0.1 | 70.9±0.1 | 71.7±1.1 | 72.0±0.2 | 72.3±0.1 | 72.7±0.3 |
| EuroSAT (Helber et al., 2019) | 93.7±0.1 | 95.1±0.1 | 95.9±0.2 | 94.4±0.1 | 95.1±0.1 | 95.3±0.1 |
| Resics45 (Cheng et al., 2017) | 75.8±0.0 | 82.5±0.2 | 84.8±0.2 | 78.4±0.0 | 81.5±0.1 | 85.9±0.0 |
| Patch Camelyon (Veeling et al., 2018) | 81.7±0.1 | 82.5±0.7 | 84.0±0.5 | 77.0±1.0 | 81.3±0.6 | 69.3±0.8 |
| Retinopathy (Kaggle & EyePacs, 2015) | 49.5±0.2 | 66.2±0.5 | 76.8±0.1 | 75.0±0.4 | 75.2±0.6 | 77.2±0.6 |
| CLEVR-count (Johnson et al., 2017) | 38.1±0.1 | 85.6±0.9 | 94.2±0. | 48.3±0.8 | 68.1±2.8 | 54.6±7.1 |
| CLEVR-dist (Johnson et al., 2017) | 33.6±0.1 | 56.1±0.8 | 57.2±2.7 | 34.8±1.5 | 47.7±1.2 | 47.9±0.8 |
| dSprites-loc (Matthey et al., 2017) | 21.6±1.1 | 74.8±1.4 | 91.6±1.9 | 12.7±1.4 | 49.7±0.6 | 91.6±1.1 |
| dSprites-ori (Matthey et al., 2017) | 43.2±1.9 | 51.3±0.7 | 62.6±0.7 | 31.7±0.8 | 52.7±0.7 | 65.9±0.3 |
| SmallNORB-azi (LeCun et al., 2004) | 11.9±0.0 | 16.2±0.1 | 19.5±0.2 | 11.5±0.9 | 15.0±0.2 | 18.7±0.2 |
| SmallNORB-elev (LeCun et al., 2004) | 28.1±0.0 | 37.0±0.6 | 44.2±0.6 | 29.3±1.9 | 34.3±0.7 | 25.8±0.9 |
| DMLab (Beattie et al., 2016) | 34.3±0.0 | 41.6±0.6 | 46.9±0.8 | 37.3±1.3 | 41.8±0.7 | 47.1±0.1 |
| KITTI-dist (Geiger et al., 2013) | 66.0±0.0 | 77.7±0.8 | 82.9±0.2 | 69.2±0.7 | 78.1±1.5 | 80.1±0.9 |
| All | 56.5 | 69.3 | 72.6 | 57.5 | 65.3 | 68.3 |
| Natural | 70.8 | 78.7 | 77.0 | 70.6 | 74.2 | 77.0 |
| Specialized | 75.2 | 81.5 | 85.3 | 81.2 | 83.3 | 81.9 |
| Structured | 34.6 | 55.0 | 62.4 | 34.4 | 48.4 | 54.0 |

## A.6 Discussion of the empirical results

FiLM works well for low shots per class because it has a small number of parameters to adapt – this gets you a long way with large pre-trained backbones – and it also prevents over-fitting as compared to adapting all the parameters in the backbone. Evidence: eventually as dataset size increases, full body adaptation is at least as good as FiLM. For larger models, this transition happens at a larger number of data points and so we expect adapters to be especially useful as the field transitions to using large foundation models. Fig. 3 provides an empirical justification of this argument, showing that FiT-LDA outperforms BiT at low-shot.

The linear head is more flexible and works best of all methods in high data, but in the low- to medium-shot setting the linear head appears to lead to increased overfitting and the meta-trained LDA head performs best in this region. See Fig. 3 for empirical justification.

The FiT approach suffers when there are a fairly large number of data points and the dataset is very far from the pretraining data (ImageNet) e.g. see the DSprites results in Table 3 where BiT beats FiT-LDA by a large margin. In addition, the LDA and QDA variants of the Naïve Bayes head are more computationally expensive compared to a linear head since LDA and QDA require inverting $d_b \times d_b$ matrices for each training iteration.

## A.7 FiLM Layer Parameter Variation

Fig. A.2a and Fig. A.2b show the magnitude of the FiLM parameters as a function of layer for FiT-LDA on CIFAR100 and SHVN, respectively, in the VTAB-1k setting. We see that for a dataset that differs from the pretraining data (SHVN), the FiLM layers are required to learn a greater degree of adjustment compared to a dataset that is similar to the pretraining data (CIFAR100). https://eprint.iacr.org/2017/715.pdf

## A.8 Additional Few-shot Federated Learning Results

Table A.6: Few-shot Federated Learning Results on CIFAR100 for different numbers of clients and shots per client. Accuracy figures are percentages and the $\pm$ sign indicates the $95\%$ confidence interval over 3 runs. Global stands for the global setting, while Personalized stands for the personalized scenario. FL indicates Federated Learning training.

| Clients | Shot | Global | | | Personalized | | |
|---|---|---|---|---|---|---|---|
| | | Lower Bound | FL | Upper Bound | Lower Bound | FL | Upper Bound |
| 10 | 2 | 42.6±1.9 | 49.3±1.3 | 52.5±0.9 | 73.2±1.4 | 80.1±0.8 | 81.7±1.2 |
| | 5 | 55.6±1.6 | 64.1±1.6 | 68.7±0.7 | 82.6±0.4 | 88.8±0.2 | 90.6±0.4 |
| | 10 | 60.7±1.2 | 71.5±0.8 | 74.7±0.3 | 86.6±0.2 | 91.8±0.5 | 92.6±0.6 |
| 50 | 2 | 59.9±0.6 | 68.5±0.3 | 73.2±0.2 | 75.4±0.9 | 84.6±0.5 | 93.4±0.2 |
| | 5 | 63.0±0.8 | 73.8±0.7 | 78.0±0.1 | 83.5±0.5 | 91.0±0.4 | 94.9±0.2 |
| | 10 | 65.8±1.1 | 77.1±0.3 | 79.2±0.3 | 87.4±0.6 | 93.4±0.3 | 95.2±0.1 |
| 100 | 2 | 65.6±0.6 | 74.4±0.3 | 77.5±0.2 | 75.7±0.9 | 85.3±0.2 | 94.6±0.2 |
| | 5 | 66.0±0.3 | 76.5±0.2 | 79.5±0.2 | 83.1±0.9 | 91.3±0.2 | 95.3±0.1 |
| | 10 | 66.8±0.3 | 78.4±0.1 | 80.2±0.1 | 87.4±0.5 | 93.6±0.2 | 95.5±0.1 |
| 500 | 2 | 68.8±0.1 | 77.9±0.4 | 80.1±0.3 | 75.5±0.2 | 85.8±0.2 | 95.6±0.1 |
| | 5 | 67.6±0.1 | 77.6±0.1 | 80.6±0.2 | 83.2±0.3 | 91.0±0.2 | 95.7±0.1 |
| | 10 | 67.7±0.1 | 79.2±0.3 | 80.8±0.2 | 87.7±0.1 | 94.3±0.1 | 96.2±0.1 |

Table A.6 shows the tabular version of Fig. 4. For the Federated Learning results, it includes only the resulting accuracy after training for 60 communication rounds. Refer to Section 4.4 for analysis.

In general, FiT can operate well with other FL aggregation methods, as they are agnostic to the neural network architectures used to learn the parameters. However, as we are constructing a Naïve Bayes head instead of training a linear head via SGD, some of the theoretical convergence results of the methods mentioned may not hold. Despite this fact, FiT has shown to have quite fast convergence in practice. In Table A.7 we provide the results of training FiT with Federated Averaging (FedAvg) and FedProx(Li et al., 2020).

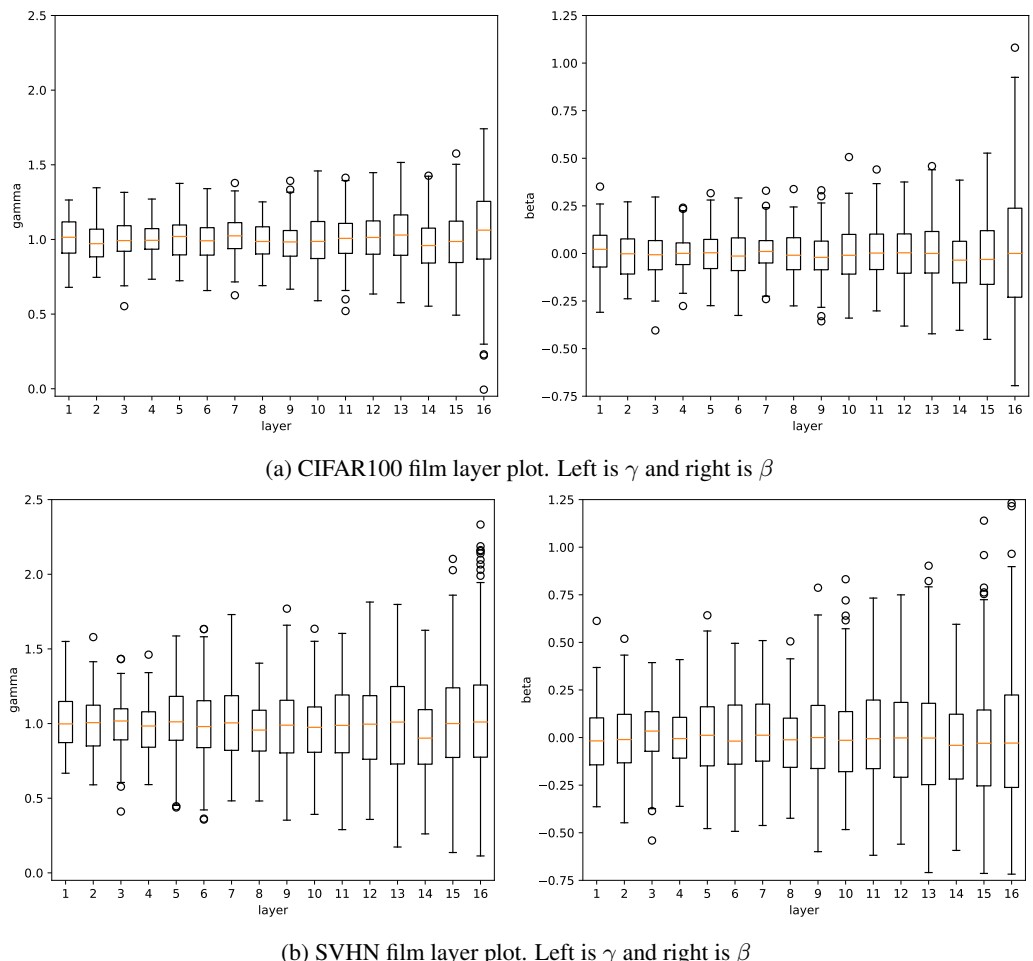

(a) CIFAR100 film layer plot. Left is $\gamma$ and right is $\beta$

(b) SVHN film layer plot. Left is $\gamma$ and right is $\beta$

Figure A.2: (Top) Box plots of the film parameter magnitudes as a function of layer in the feature extractor.

To test distributed training of a FIT model on a more extreme, non-natural image dataset, we also include the results for federated training of FIT on the Quickdraw dataset. As there is no pre-defined train/test split for the Quickdraw dataset, we randomly choose 100 samples from each of the 345 classes and use them for testing. We train all federated training models for 120 communication rounds, with 5 clients per round, and 10 update steps per client. Since Quickdraw is a more difficult dataset than CIFAR100, it requires more communication rounds for training. Each client has 35 classes, which are sampled randomly at the start of training. In our experiments, we omit the 10-clients case, as the overall amount of data in the system is not enough to even train a robust global upper bound baseline model.

Fig. A.3 shows global and personalized classification accuracy as a function of communication cost for different numbers of clients and shots per client for Quickdraw, while Table A.9 shows the tabular version of this figure.

Similarly to CIFAR100 experiments, we compare FIT to BiT on the Quickdraw dataset. We train both models for 60 communication rounds. The results are presented in Table A.8. For 5 and 10 shots per class, FIT performs similarly to BiT for all numbers of clients tested. However, for 2-shot, BiT outperforms FIT for all numbers of clients tested, but the performance gap is reduced as the number of clients is increased. We attribute this behavior to the use of the ProtoNets classifier. During training at each client's optimization step, we construct a N-way local ProtoNets classifier using only one image per class, which may result in a classifier not robust enough for reasonable optimization of the FiLM layers. This hypothesis is also supported by Fig. A.3, where we see a huge gap between

Table A.7: FIT trained with different FL algorithms on CIFAR100 for different numbers of clients and shots per client. Federated averaging (FedAvg) and FedProx methods were used. For FedProx $\mu = 0.01$ is used. Accuracy figures are percentages and the $\pm$ sign indicates the $95\%$ confidence interval over 3 runs. Global stands for the global setting, while Personalized stands for the personalized scenario.

| | | Global | | Personalized | |
|---|---|---|---|---|---|
| Clients | Shot | FedProx | FedAvg | FedProx | FedAvg |
| | 2 | 50.1±2.2 | 49.3±1.3 | 80.5±1.1 | 80.1±0.8 |
| 10 | 5 | 64.3±1.6 | 64.1±1.6 | 88.9±0.3 | 88.8±0.2 |
| | 10 | 71.5±0.8 | 71.5±0.8 | 91.9±0.3 | 91.8±0.5 |
| | 2 | 68.8±0.5 | 68.5±0.3 | 84.8±0.6 | 84.6±0.5 |
| 50 | 5 | 74.0±0.5 | 73.8±0.7 | 91.2±0.4 | 91.0±0.4 |
| | 10 | 77.1±0.1 | 77.1±0.3 | 93.5±0.2 | 93.4±0.3 |
| | 2 | 74.5±0.2 | 74.4±0.3 | 85.3±0.2 | 85.3±0.2 |
| 100 | 5 | 76.9±0.3 | 76.5±0.2 | 91.4±0.3 | 91.3±0.2 |
| | 10 | 78.4±0.2 | 78.4±0.1 | 93.7± 0.2 | 93.6±0.2 |
| | 2 | 78.1±0.4 | 77.9±0.4 | 86.1±0.3 | 85.8±0.2 |
| 500 | 5 | 77.9±0.1 | 77.6±0.1 | 91.1±0.2 | 91.0±0.2 |
| | 10 | 79.3±0.3 | 79.2±0.3 | 94.3±0.1 | 94.3±0.1 |

Table A.8: Comparison of BiT and FIT in few-shot federated setting on Quickdraw for different numbers of clients and shots per client. Global setting is used. Accuracy figures are percentages and the $\pm$ sign indicates the $95\%$ confidence interval over 3 runs. Both models were trained for 60 communication rounds. Parameter cost indicates the number of parameters sent in server-client communication. *Per round* is the number of parameters transmitted in each communication round and *Overall* is the number of parameters transmitted during the whole training.

| | Parameter cost | | 50 Clients | | | 100 Clients | | | 500 Clients | | |
|---|---|---|---|---|---|---|---|---|---|---|---|
| Models | Per round | Overall | 2-Shot | 5-Shot | 10-Shot | 2-Shot | 5-Shot | 10-Shot | 2-Shot | 5-Shot | 10-Shot |
| FIT | **0.1M** | **7M** | 26.6±1.0 | 40.8±0.6 | 45.5±0.2 | 32.0±2.0 | 44.0±0.7 | 46.6±0.2 | 39.4±1.9 | 45.6±0.1 | 47.9±0.4 |
| BiT | 242M | 14.5B | **37.5±2.2** | 42.4±1.7 | 44.4±0.7 | **40.4±0.5** | 44.2±1.4 | 46.4±0.9 | 41.1±2.5 | 45.7±2.6 | 47.5±1.7 |

the performance of Federated Learning and centralized learning (the upper bound) for 2-shot. In contrast, the linear head in BiT is not local and is trained using all clients' data, thus avoiding this pitfall. This leads us to the following observation – if there is enough local client data to construct a robust metric-based classifier, then Naïve Bayes head helps to significantly reduce communication cost without sacrificing the final model quality. However, if there is not enough local client data, then the use of a linear classification head may be more appropriate.

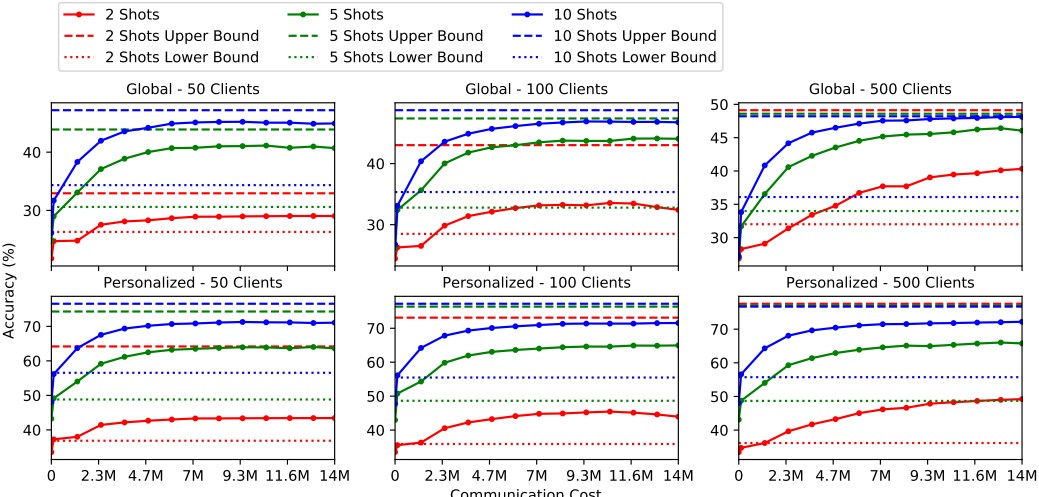

Figure A.3: Global and personalized classification accuracy as a function of communication cost over 120 rounds for different numbers of clients and shots per client on Quickdraw. Classification accuracy is on the vertical axis and is the average of 3 runs with different data sampling seeds. The color of the line indicates the number of shots per class. The solid line shows the federated learning model, while dashed and dotted lines indicate the upper and lower bounds baselines, respectively.

Table A.9: Few-shot Federated Learning Results on Quickdraw for different numbers of clients and shots per client. Accuracy figures are percentages and the $\pm$ sign indicates the $95\%$ confidence interval over 3 runs. Global stands for the global setting, while Personalized stands for the personalized scenario. FL indicates Federated Learning training.

| Clients | Shot | Global | | | Personalized | | |
|---|---|---|---|---|---|---|---|
| | | Lower Bound | FL | Upper Bound | Lower Bound | FL | Upper Bound |
| 50 | 2 | 26.3±0.4 | 29.0±0.5 | 32.9±0.6 | 36.9±0.2 | 43.4±0.6 | 64.2±0.8 |
| | 5 | 30.6±0.3 | 40.7±0.6 | 43.9±0.5 | 48.8±0.4 | 63.6±0.3 | 74.3±0.4 |
| | 10 | 34.3±0.7 | 44.9±0.1 | 47.2±0.2 | 56.5±0.8 | 71.1±0.1 | 76.6±0.2 |
| 100 | 2 | 28.5±0.1 | 32.4±0.8 | 43.0±1.0 | 35.9±0.3 | 43.9±1.0 | 73.1±0.4 |
| | 5 | 32.8±0.4 | 44.1±0.4 | 47.4±0.1 | 48.6±0.9 | 64.9±0.8 | 76.4±0.3 |
| | 10 | 35.3±0.1 | 46.8±0.1 | 48.7±0.2 | 55.5±0.4 | 71.5±0.5 | 77.2±0.3 |
| 500 | 2 | 32.0±0.3 | 40.3±2.4 | 49.1±0.3 | 36.2±0.1 | 49.2±2.4 | 77.6±0.3 |
| | 5 | 34.0±0.1 | 46.1±0.1 | 48.6±2.2 | 48.7±0.2 | 65.8±0.3 | 77.0±1.7 |
| | 10 | 36.1±0.1 | 48.1±0.4 | 48.2±1.2 | 55.7±0.1 | 72.2±0.2 | 76.7±0.9 |

## A.9 CONNECTIONS TO PERSONALIZED FEDERATED LEARNING

Partial model based personalized federated learning (PFL) (Collins et al., 2021; Arivazhagan et al., 2019; Liang et al., 2020; Pillutla et al., 2022) is related to the personalized setting in our federated learning experiments, but it is different in important regards. With regard to the similarities, the local Naïve Bayes head parameters in our approach may be considered as "personal" parameters (there are personalized parameters for each client), while the FiLM parameters could be viewed as "shared" parameters. However, there are a few major differences between the ideas:

- The personalized parameters in our setting, i.e. the ProtoNets head, does not require an optimization loop to be learned. This simplifies deployment significantly.
- The partial model personalization literature (Collins et al., 2021; Arivazhagan et al., 2019; Liang et al., 2020; Pillutla et al., 2022) is more concerned with proposing stable federated training algorithms that would work within heterogeneous settings, where clients have diverse data and standard FedAvg algorithm would fail, thus necessitating a need to introduce 'personal' parameters. In contrast, personalized heads are required in our setup as each user has a different classification task to perform. Moreover, we propose a particular architecture that would be highly suitable for federated learning applications, as the number of parameters required to be transmitted is small, while most methods in PFL propose architecture agnostic optimization algorithms. Our model can then be trained with an arbitrary Federated Learning algorithm.
- Another distinctive difference between the model personalization federated learning literature and our work is that in the former most of the methods train all model's parameters, while in our work we used a deep pretrained network and fine-tune only the FiLM layers.

## A.10 FIT TRAINING ALGORITHMS

Algorithm A.1 and Algorithm A.2 detail how episodic fine-tuning tasks are split and sampled, respectively, for use in the FIT training protocol.

---

**Algorithm A.1** Splitting the downstream dataset $\mathcal{D}$

---

**Require:** $\mathcal{D} = \{(\boldsymbol{x}_n, y_n)\}_{n=1}^N = \{\boldsymbol{x}, \boldsymbol{y}\}$: downstream dataset
**Require:** `unique()` $\equiv$ function that returns a list of unique classes and list of counts of each class
**Require:** `select_by_class()` $\equiv$ function that extracts samples of a specified class from a dataset
1: **procedure** SPLIT($\mathcal{D}$)
2:     $\mathcal{D}_{train} \leftarrow []$                                              ▷ Create an empty list to hold $\mathcal{D}_{train}$
3:     $\mathcal{D}_{test} \leftarrow []$                                              ▷ Create an empty list to hold $\mathcal{D}_{test}$
4:     `classes, class_counts` $\leftarrow$ `unique`($\boldsymbol{y}$)
5:     **for all** $c \in$ `classes` **do**
6:         `assert(class_counts`($c$) $> 1$)            ▷ Require a minimum of 2 shots per class.
7:         `train_count` $\leftarrow$ `ceil(class_counts`($c$)$/2$)
8:         $\mathcal{D}_c \leftarrow$ `select_by_class`($c$)                    ▷ Select examples of class $c$ from $\mathcal{D}$
9:         $\mathcal{D}_{train} \leftarrow \mathcal{D}_{train} + \mathcal{D}_c[:$ `train_count`$]$   ▷ Add `train_count` examples to $\mathcal{D}_{train}$
10:         $\mathcal{D}_{test} \leftarrow \mathcal{D}_{test} + \mathcal{D}_c[$`train_count` $:]$          ▷ Add remaining examples to $\mathcal{D}_{test}$
11:     **end for**
12:     **return** $\mathcal{D}_{train}, \mathcal{D}_{test}$
13: **end procedure**

---

## A.11 TRAINING AND EVALUATION DETAILS

In this section, we provide implementation details for all of the experiments in Section 4.

### A.11.1 FEW-SHOT AND VTAB-1K TRANSFER LEARNING EXPERIMENTS

**FIT** All of the FIT few-shot and VTAB-1k transfer learning experiments were carried out on a single NVIDIA A100 GPU with 80GB of memory. The Adam optimizer (Kingma & Ba, 2015) with a constant learning rate of 0.0035, for 400 iterations, and $|\mathcal{D}_S^\tau|$=100 was used throughout. No data augmentation was used and images were scaled to 384×384 pixels unless the image size was 32×32 pixels or less, in which case the images were scaled to 224×224 pixels. In each iteration of episodic fine tuning, we perform a single step of gradient ascent on $\mathcal{D}_Q^\tau$. These hyper-parameters were empirically derived from a small number of runs.

FIT-QDA, FIT-LDA, and FIT-ProtoNets take approximately 12, 10, and 9 hours, respectively, to fine-tune on all 19 VTAB datasets and 5, 3, and 3 hours, respectively, to fine tune all shots on the 4 low-shot datasets.

For the FIT-LDA results using the EfficientNetV2-M backbone in Table 3, we used 1000 iterations instead of 400 and ran the experiments on 4 NVIDIA A100 GPUs, each with 80GB of memory.

---

**Algorithm A.2** Sampling a task $\tau$

---

**Require:** $\mathcal{D}_{train} = \{(\boldsymbol{x}_s, y_s)\}_{s=1}^{S_\tau} = \{\boldsymbol{x}_S, \boldsymbol{y}_S\}$: train portion of downstream dataset
**Require:** $\mathcal{D}_{test} = \{(\boldsymbol{x}_q, y_q)\}_{q=1}^{Q_\tau} = \{\boldsymbol{x}_Q, \boldsymbol{y}_Q\}$: test portion of downstream dataset
**Require:** `support_set_size`: size of the support set $|\mathcal{D}_S^\tau|$
**Require:** `unique()` $\equiv$ function that returns a list of unique classes and list of counts of each class
**Require:** `randint`$(min, max)$ $\equiv$ function that returns a random integer between $min$ and $max$
**Require:** `choice`$(range, count)$ $\equiv$ function that returns a random list of $count$ integers from $range$

1: **procedure** SAMPLE_TASK($\mathcal{D}_{train}, \mathcal{D}_{test}$, `support_set_size`)
2:      $\mathcal{D}_S^\tau \leftarrow []$                                      ▷ Create an empty list to hold $\mathcal{D}_S^\tau$
3:      $\mathcal{D}_Q^\tau \leftarrow []$                                      ▷ Create an empty list to hold $\mathcal{D}_Q^\tau$
4:      `train_classes, train_class_counts` $\leftarrow$ `unique`($\boldsymbol{y}_S$)
5:      `test_classes, test_class_counts` $\leftarrow$ `unique`($\boldsymbol{y}_Q$)
6:      `min_way` $\leftarrow$ `min`(`len`(`train_classes`), 5)
7:      `max_way` $\leftarrow$ `min`(`len`(`train_classes`), `support_set_size`)
8:      `way` $\leftarrow$ `randint`(`min_way, max_way`)         ▷ Classification way to use for this task
9:      `selected_classes` $\leftarrow$ `choice`(`train_classes, way`)     ▷ List of classes to use in this task
10:      `balanced_shots` = `max`(`round`(`support_set_size` / `len`(`selected_classes`)), 1)
11:      `max_test_shots` $\leftarrow$ `max`(1, `floor`(2000/`way`))
12:      **for all** $c \in$ `selected_classes` **do**
13:          `class_shots` $\leftarrow$ `train_class_counts`($c$)
14:          `shots_to_use` $\leftarrow$ `min`(`class_shots, balanced_shots`)
15:          `selected_shots` $\leftarrow$ `choice`(`class_shots, shots_to_use`) ▷ Support shot list
16:          $\mathcal{D}_S^\tau \leftarrow \mathcal{D}_S^\tau + \mathcal{D}_{train}[\text{selected\_shots}]$             ▷ Add examples to $\mathcal{D}_S^\tau$
17:          `class_shots` $\leftarrow$ `test_class_counts`($c$)
18:          `shots_to_use` $\leftarrow$ `min`(`class_shots, max_test_shots`)
19:          `selected_shots` $\leftarrow$ `choice`(`class_shots, shots_to_use`)    ▷ Query shot list
20:          $\mathcal{D}_Q^\tau \leftarrow \mathcal{D}_Q^\tau + \mathcal{D}_{test}[\text{selected\_shots}]$            ▷ Add examples to $\mathcal{D}_Q^\tau$
21:      **end for**
22:      **return** $\mathcal{D}_S^\tau, \mathcal{D}_Q^\tau$
23: **end procedure**

---

**BiT** For the BiT few-shot experiments, we used the code supplied by the authors (Kolesnikov et al., 2020) with minor augmentations to read additional datasets. The BiT few-shot experiments were run on a single NVIDIA V100 GPU with 16GB.

For the BiT VTAB-1k experiments, we used the three fine-tuned models for each of the datasets that were provided by the authors (Kolesnikov et al., 2020). We evaluated all of the models on the respective test splits for each dataset and averaged the results of the three models. The BiT-HyperRule (Kolesnikov et al., 2019) was respected in all runs. These experiments were executed on a single NVIDIA GeForce RTX 3090 with 24GB of memory.

### A.11.2 PERSONALIZATION ON ORBIT EXPERIMENTS

The personalization experiments were carried out on a single NVIDIA GeForce RTX 3090 with 24GB of memory. It takes approximately 5 hours to train FIT-LDA personalization models for all the ORBIT (Massiceti et al., 2021) test tasks. We derived all hyperparameters empirically from a small number of runs. We used the ORBIT codebase[1] in our experiments, only adding the code for splitting test user tasks and slightly modifying the main training loop to make it suitable for FIT training.

In the comparison of FIT to standard benchmarks on ORBIT (upper part of Table 4), all methods use an EfficientNet-B0 ($d_b = 1280$) as the feature extractor and an image size of $224 \times 224$. FIT-LDA, FineTuner (Yosinski et al., 2014) and Simple CNAPs (Bateni et al., 2020) use a backbone pretrained on ImageNet (Deng et al., 2009), while ProtoNets (Snell et al., 2017) meta-trained the weights of the feature extractor on Meta-Dataset (Dumoulin et al., 2021). The task encoder in Simple CNAPs (Bateni et al., 2020) is meta-trained on Meta-Dataset. The results for all models (FineTuner (Yosinski et al., 2014), Simple CNAPs (Bateni et al., 2020) and ProtoNets (Snell et al., 2017)) are from (Bronskill et al., 2021). FiLM layers in FIT-LDA are added to the feature extractor as described in Section 2, resulting in $|\psi| = 20544$.

For the comparison of FIT and BiT (lower part of Table 4), we use the BiT-M-R50x1 backbone pretrained on ImageNet-21K. BiT was trained using the BiT-HyperRule (Kolesnikov et al., 2019) on every user task. For experiments with BiT, we used the code supplied by the authors (Kolesnikov et al., 2019) with minor augmentations to read the ORBIT dataset.

We follow the task sampling protocols described in (Massiceti et al., 2021), and train the FIT model for 50 optimization steps using the Adam optimizer with a learning rate of 0.007 for EfficientNet-B0 and 0.003 for BiT-M-R50x1. The ORBIT test tasks have a slightly different structure in comparison to standard few-shot classification tasks, so in Algorithm A.3 we provide a modified version of the data splitting for the classifier head construction. In particular, each test user has a number of objects (classes) they want to recognize, with several videos recorded per object. Each video is split into clips, consecutive 8-frame parts of the video. A user test task is comprised of these clips, randomly sampled from different videos of the user's objects, and associated labels. Since clips sampled from the same video can be semantically similar, we split the test task so that clips from the same video can only be in either the support or query set, except for the cases when there is only one video of an object available.

### A.11.3 FEDERATED LEARNING EXPERIMENTS

For each local update a new Adam optimizer is initialized. In each communication round, 5 clients are randomly chosen for making model updates. All of the federated learning experiments were carried out on a single NVIDIA A100 GPU with 80GB of memory. In all experiments we use FIT with the BiT-M-R50x1 (Kolesnikov et al., 2019) backbone pretrained on the ImageNet-21K (Russakovsky et al., 2015) dataset and a ProtoNets head. We derive all hyperparameters empirically from a small number of runs.

**BiT** As the training protocol proposed in BiT cannot be directly applied to the federated learning setting, we simplify it and train the BiT model for 60 communication rounds with a constant learning rate of 0.003.

---

[1] https://github.com/microsoft/ORBIT-Dataset

---

**Algorithm A.3** Splitting a test task $\mathcal{D}$ for ORBIT personalization experiments

---

**Require:** $\mathcal{D}$: downstream dataset; $\mathcal{D} = \{\mathcal{D}_c\}_{c=1}^C$, where $C$ is the number of classes in test task, $\mathcal{D}_c$ is data of class $c$; $\mathcal{D}_c = \{V_{ci}\}_{i=1}^{n_c}$, where $n_c$ is the number of videos in class $c$, $V_{ci}$ is the set of clips from $i$th video of class $c$; $V_{ci} = \{(x_{cij}, c)\}_{j=1}^{n_{ci}}$, where $n_{ci}$ is the number of clips in $i$th video of class $c$, $x_{cij}$ is the $j$th clip from video $V_{ci}$
**Require:** `batch_size`: size of context split
**Require:** `choose`$(n, m) \equiv$ function that randomly samples $m$ different integers from a set $\{i\}_{i=1}^n$
**Require:** `select_by_index`$(\mathcal{D}, i) \equiv$ function that extracts samples of indices $i$ from a dataset $\mathcal{D}$
**Require:** `diff`$(a, b) \equiv$ function that computes set difference between sets $a$ and $b$
**Require:** `range`$(n) \equiv$ function that returns a set of values $\{i\}_{i=1}^n$

  1: **procedure** SPLIT_ORBIT_TASK($\mathcal{D}$)
  2:     $\mathcal{D}_{train} \leftarrow []$
  3:     $\mathcal{D}_{test} \leftarrow []$
  4:     `num_clips` $\leftarrow$ `floor`$(\texttt{batch\_size}/C)$
  5:     **for** $c \leftarrow 1$ to $C$ **do**
  6:         `num_context_videos` $\leftarrow$ `ceil`$(n_c/2)$
  7:         `context_videos_indices` $\leftarrow$ `choose`$(n_c, \texttt{num\_context\_videos})$
  8:         `num_clips_per_video` $\leftarrow$ `floor`$(\texttt{num\_clips}/\texttt{num\_context\_videos})$
  9:         **if** $n_c = 1$ **then**
10:             `context_clips_indices` $\leftarrow$ `choose`$(n_{c1}, \texttt{num\_clips\_per\_video})$
11:             `target_clips_indices` $\leftarrow$ `diff`$(\texttt{range}(n_{c1}), \texttt{context\_clips\_indices})$
12:             $\mathcal{D}_{train} \leftarrow \mathcal{D}_{train} + $ `select_by_index`$(V_{c1}, \texttt{context\_clips\_indices})$
13:             $\mathcal{D}_{test} \leftarrow \mathcal{D}_{test} + $ `select_by_index`$(V_{c1}, \texttt{target\_clips\_indices})$
14:         **else**
15:             **for** $j \leftarrow$ `context_videos_indices` **do**
16:                 `context_clips_indices` $\leftarrow$ `choose`$(n_{cj}, \texttt{num\_clips\_per\_video})$
17:                 $\mathcal{D}_{train} \leftarrow \mathcal{D}_{train} + $ `select_by_index`$(V_{cj}, \texttt{context\_clips\_indices})$
18:             **end for**
19:             **for** $j \leftarrow$ `diff`$(\texttt{range}(n_c), \texttt{context\_videos\_indices})$ **do**
20:                 $\mathcal{D}_{test} \leftarrow \mathcal{D}_{test} + V_{cj}$
21:             **end for**
22:         **end if**
23:     **end for**
24:     **return** $\mathcal{D}_{train}, \mathcal{D}_{test}$
25: **end procedure**

---

**CIFAR100**   We train all federated learning models with different number of clients and shots per client for 60 communication rounds, with 5 clients per round and 10 update steps per client. Each client has 10 classes, which are sampled uniformly before the start of training. In our experiments, we used 2, 5 or 10 shot per class. For the global setting, we used the CIFAR100 test split for the global model evaluation. For the personalized setting, we test each client using the CIFAR100 test split, but using only the classes that a particular client owns. This results in 10-class classification task. We then report the average accuracy across all clients. We use a learning rate of 0.003 at the start of the training, decaying it by 0.3 every 20 communication rounds. Upper and lower bound baselines for both the global and personalized scenarios were trained for 400 epochs using the Adam optimizer with a constant learning rate of 0.003. It takes around 20 minutes to train federated learning models, with slightly more training time required for the models with a larger number of shots.

**Quickdraw**   Each client has 35 classes, which are sampled uniformly at the start of training. As there is no pre-defined train/test split for the Quickdraw dataset, we randomly choose 100 samples from each of the 345 classes and use them for testing. This test data is then used to test both the personalized and global settings as described in the paragraph above. We train all federated training models for 120 communication rounds, with 5 clients per round, and 10 update steps per client. We use a constant learning rate of 0.006 for training all federated learning models. Upper bound baseline models, which require training a global model using all available data, were trained for 3000 steps using the Adam optimizer with a constant learning rate of 0.006. Lower baseline models, requiring training a personalized model for each individual, were trained for 400 steps using the Adam optimizer with a learning rate of 0.006. It takes around 2 hours to train federated learning models, with slightly more training time required for the models with a larger number of shots.

