# OpenReview forum: "FiT: Parameter Efficient Few-shot Transfer Learning for Personalized and Federated Image Classification"
_ICLR.cc/2023/Conference — ICLR 2023 poster_

### Official Review · Reviewer_8TEt · 2022-10-24

**Confidence:** 4
**Correctness:** 3
**Technical Novelty And Significance:** 4
**Empirical Novelty And Significance:** 3
**Recommendation:** 6

**Clarity, Quality, Novelty And Reproducibility:**

  The quality of the paper is relatively high, but the methods section is not clear enough, it is slightly difficult to read, and it is  original.

**Strength And Weaknesses:**

  Strength: The method proposed in this paper is simple, efficient and easy to implement. The adequate experimental results on few-shot,VTAB-1k benchmark, Personalization video dataset and Few-shot Federated Learning verify the efficiency of FiT.


Weaknesses: The motivation for using a Naive Bayes classifier is not clear, why it would be better than a traditional MLP classification layer.The design form of Feature-wise Linear Modulation (FiLM) layers is very simple and direct. Why is such design effective?


**Summary Of The Paper:**

   The paper proposed a parameter and data efficient architecture for low-shot transfer learning consisting of an automatically configured Naive Bayes classifier and FiLM layers that are used to adapt a fixed, pretrained backbone to a downstream dataset.

**Summary Of The Review:**

   This paper proposed Feature-wise Linear Modulation layers, which can be easily inserted into Resnet or Efficient networks, and only a few parameters are added. The use of an automatically configured Naive Bayes classifier to replace the traditional classification layer significantly improves the performance of the few shot and image/video classification.

---

> ### Author Response · Authors · 2022-11-11
> **Response to Reviewer 8TEt**
>
> Thanks for taking the time to review our paper. Responses to your questions and issues are below:
>
> **The motivation for using a Naive Bayes classifier is not clear, why it would be better than a traditional MLP classification layer.**
>
> The linear head is more flexible and works best of all methods in the high data regime, but in the low- to medium-shot setting the linear head appears to lead to increased overfitting and our results show that the meta-trained Naïve Bayes head performs best in this region. We also note that the Naïve-Bayes-LDA head converges more quickly than the linear head (see Figure 2b).
>
> **The design form of Feature-wise Linear Modulation (FiLM) layers is very simple and direct. Why is such design effective?**
>
> FiLM layers are a proven and simple approach to adapting pretrained bodies to new data (for example, see Requeima et al., Fast and flexible multi-task classification using conditional neural adaptive processes. In Proceedings of the 33rd Annual Conference on Neural Information Processing Systems (NeurIPS), pp. 7957–7968, 2019.). FiLM works well for low shots per class because it has a small number of parameters to adapt – this gets you a long way with large pre-trained backbones -- and it also prevents over-fitting as compared to adapting all the parameters in the backbone. Evidence: eventually as dataset size increases, full body adaptation wins / is as good as FiLM. For larger models, this transition happens at a larger number of data points and so we expect adapters to be especially useful as the field transitions to using large foundation models. Figure 3 provides an empirical justification of this argument, showing that FiT-LDA outperforms BiT at low-shot.
>
> **The quality of the paper is relatively high, but the methods section is not clear enough, it is slightly difficult to read, and it is original.**
>
> Thanks for the feedback, we aim to improve the readability based on the reviewers’ feedback and update the submission.

---

### Official Review · Reviewer_YJfE · 2022-10-27

**Confidence:** 2
**Correctness:** 3
**Technical Novelty And Significance:** 3
**Empirical Novelty And Significance:** 3
**Recommendation:** 6

**Clarity, Quality, Novelty And Reproducibility:**

The paper is well-written, with a comparison table wrt BiT that is much appreciated. The paper is a novel combination of existing works (FiLM, [Dumoulin'2021) for applications in personalized and FL-based image classification which outperforms BiT.

**Strength And Weaknesses:**

**Strengths**
1) The proposed FiT has superior parameter efficiency and classification accuracy using orders of magnitude fewer updateable parameters and communication cost than BiT.
2) The paper is well-organized with detailed experiments and comparisons across various downstream tasks in few-shot learning.


**Weakness**
1) The paper combines ideas from FiLM (to learn from little data and generalize to more complex and/or substantially different data) layers and meta-learning-inspired episodic training protocol [Dumoulin'2021] to outperform transfer learning baseline BiT.



**Summary Of The Paper:**

The authors propose FiLM Transfer (FiT) which combines transfer learning and meta-learning. FiT aims to achieve parameter efficiency under a limited number of training samples without degrading accuracy. This potentially results in inexpensive model updates for personalization and communication-efficient federated learning.
FiT backbone is a frozen pre-trained network that is augmented with parameter-efficient FiLM layers (for adapting to the new tasks), and a Naive Bayes classifier layer with fewer updateable parameters rather than the linear layer.

**Summary Of The Review:**

The paper proposes a technique, FiT to yield parameter efficient models that outperform BiT by combining transfer learning with meta-learning.

---

> ### Author Response · Authors · 2022-11-11
> **Response to Reviewer YJfE**
>
> Thanks for taking the time to review our paper. Responses to your questions/issues are below:
>
> **The paper combines ideas from FiLM (to learn from little data and generalize to more complex and/or substantially different data) layers and meta-learning-inspired episodic training protocol [Dumoulin'2021] to outperform transfer learning baseline BiT.**
>
> We make the following novel methodological contributions:
> - The QDA/LDA-like head with meta-trained regularization parameters $e$ is a new contribution. It generalizes standard Mahalanobis distance heads by introducing the meta-trained covariance blending factors and our results demonstrate that it outperforms standard ProtoNets by a significant margin.
> - We are not aware that FiLM based fine-tuning of the body has been used before for few-shot learning, although it has been applied to high-shot settings by the transfer learning community.
> - The combination of the LDA meta-trained head and the FiLM fine-tuned body is novel and we found especially performant, both in terms of classification accuracy and parameter efficiency.

---

### Official Review · Reviewer_DFdq · 2022-11-02

**Confidence:** 3
**Correctness:** 4
**Technical Novelty And Significance:** 4
**Empirical Novelty And Significance:** 4
**Recommendation:** 8

**Clarity, Quality, Novelty And Reproducibility:**

In general the paper is well written, the authors explained in detail why they propose these techniques.
The experimental results are fully discussed.

**Strength And Weaknesses:**

The paper is well motivated as both personalization and federated learning require parameter efficient and training data efficient model.
The proposed idea is succinct and to me it has potential to be generalized to applications other than CV.
The paper provides sufficient and convincing experiment results to show the superior performance of the proposed architecture.

Some details need to clarify or fix:
1. During the episodic fine tuning, in each iteration, do you optimize \phi and e via gradient ascent for only one step, or optimize them by running the gradient ascent for multiple steps?
2. In section 4.1, for the one-shot learning, how do you split the downstream dataset D as D_train and D_test?
3. In the second paragraph on page 8, it's not clear to me why Naive Bayes head is not transferred to the server. Do you assume that there are no shared classes among clients?
4. There is a typo in the 4th paragraph on page 3, "b_d" should be "d_b"

**Summary Of The Paper:**

This paper proposes a parameter and data efficient network architecture and the corresponding training procedure for personalization and federated learning. The proposed architecture includes FiLM layers in the pretrained backbones and takes Naive Bayes classifier as the head layer.

**Summary Of The Review:**

Overall, it's a good paper, I'd suggest to accept this paper for publication.

---

> ### Author Response · Authors · 2022-11-11
> **Response to Reviewer DFdq**
>
> Thanks for taking the time to review our paper. Responses to your questions and issues are below:
>
> **During the episodic fine tuning, in each iteration, do you optimize \phi and e via gradient ascent for only one step, or optimize them by running the gradient ascent for multiple steps?**
>
> In each iteration of episodic fine-tuning, we take only one gradient ascent step. We have added this information to section A.11.1 where the specifics of the training protocol are detailed.
>
> **In section 4.1, for the one-shot learning, how do you split the downstream dataset D as D_train and D_test?**
>
> When there is only one shot per class, it is not possible to split dataset $\mathcal{D}$. In this case, we forgo episodic fine-tuning and simply use the default values for the FiLM layer parameters (i.e. $\gamma = 1$ and $\beta = 0$) and $e=(0.5, 0.5, 1.0)$ instead of learning them and then make predictions on the test set. Surprisingly, this works extremely well (refer to Figure 3 and Table A.1). We have updated the last two sentences in the FiT Training paragraph of Section 2 to clarify this.
>
> **In the second paragraph on page 8, it's not clear to me why Naive Bayes head is not transferred to the server. Do you assume that there are no shared classes among clients?**
>
> You could transfer the Naïve Bayes head in the way you suggest. However, this has a large communication cost as there are a large number of parameters in the classification head due to the computed means. Transferring these parameters would have the same computational cost as transferring linear classification layer, thus making our approach less effective for Federated Learning.  Thankfully, we show that high performance can be achieved without communicating the head. Specifically, during training, we construct and use a local Naïve Bayes head for each client independently. For evaluation in the personalized scenario, we test the final model with only a local Naïve Bayes head. For evaluation in the global scenario, after the final training step, all clients send their local heads for the construction of one global head.
>
> **There is a typo in the 4th paragraph on page 3, "b_d" should be "d_b"**
>
> Thanks for spotting this. It is now fixed in the revised version of the paper.

---

### Official Review · Reviewer_BNuH · 2022-11-03

**Confidence:** 4
**Correctness:** 3
**Technical Novelty And Significance:** 2
**Empirical Novelty And Significance:** 3
**Recommendation:** 6

**Clarity, Quality, Novelty And Reproducibility:**

The paper is easy to follow, but the originality and novelty are a bit incremental and limited to some extent.

**Strength And Weaknesses:**

Strengths:
1. The motivation is well-elaborated with regard to the limited data and communication efficiency.
2. The writing is easy to follow.

Weakness:
1. The design of the network architectures are basically based on the ideas from the transfer learning (frozen backbone) and meta learning (Naive Bayes final layer). Is the any difference of the design in FIT when compared with the original ideas in transfer learning and meta learning? or just simply using the two ideas without modification? Please clarify it.
2. There are two personalization concepts in this work. One is in the evaluation on the ORBIT dataset. Another one is in the federated learning. What is the difference and the relationships between these two?
4. In the experiments of few shot federated learning, the defined upper bound and lower bound for reference are useful for performance analysis during the evaluation of different settings. However, the details of model evaluation/test are not provided, for the global and personal models, for example, the setup and partitions of the datasets used in global or local test which are important in federated learning.
5. In the first sentence of the last paragraph in Page 8, the authors mentioned that, no suitable federated learning system for comparison. As the key idea of the proposed method in the scenarios of federated learning is based on partial model sharing/update, why the authors did not include the comparison of some partial model transmission-based federated learning (FL) methods or in personalized FL (PFL) scenarios, e.g., the sparisification based FL and PFL in which the communication efficiency and the performance of personal models are investigated.
6. In FL settings used on Tab. A.7 and Fig. A.3, the performance of conventional models based FL methods should be provided, with regard to accuracy and communication efficiency. It will help to demonstrate the effectiveness.
7. In the writing, more discussion should be provided around the proposed method in the federated settings, such as the advantages and the difference from the conventional partial model based PFL methods.



**Summary Of The Paper:**

This paper proposes FiLM Transfer framework (FIT) for image classification tasks via incorporating the ideas from transfer learning and meta learning. The proposed method aims to address the learning with small data and the communication efficiency in distributed learning, in the context of the personalization and federated learning. The FIT method basically consists of a pretrained backbone with an automatically configured Naive Bayes final layer classifier. The outperformances are demonstrated through the experiments in few shot, personalization and federated learning tasks.



**Summary Of The Review:**

The merits and concerns are presented in the  main review. The main concerns are about the novelty of the network design, as well as the experimental design in federated settings.

---

> ### Author Response · Authors · 2022-11-11
> **Response to Reviewer BNuH (Part 1)**
>
> Thanks for taking the time to review our paper. Responses to your questions and issues are below:
>
> **The design of the network architectures are basically based on the ideas from the transfer learning (frozen backbone) and meta learning (Naive Bayes final layer). Is the any difference of the design in FIT when compared with the original ideas in transfer learning and meta learning? or just simply using the two ideas without modification? Please clarify it.**
>
> We make the following novel methodological contributions:
>
> - The QDA/LDA-like head with meta-trained regularization parameters $e$ is a new contribution. It generalizes standard Mahalanobis distance heads by introducing the meta-trained covariance blending factors and our results demonstrate that it outperforms standard ProtoNets by a significant margin.
>
> - We are not aware that FiLM based fine-tuning of the body has been used before for few-shot learning, although it has been applied to high-shot settings by the transfer learning community.
>
> - The combination of the LDA meta-trained head and the FiLM fine-tuned body is novel and we found especially performant, both in terms of classification accuracy and parameter efficiency.
>
> **There are two personalization concepts in this work. One is in the evaluation on the ORBIT dataset. Another one is in the federated learning. What is the difference and the relationships between these two?**
>
> In the ORBIT experiments, only a single user’s data is used to train the body adapter parameters – the model is personalized to each user specifically. In the federated learning experiments, all clients’ data is used to train the body (and the head in the non-personalized setting) using the FedAvg algorithm – the model’s body is shared across all users / clients.
>
> **In the experiments of few shot federated learning, the defined upper bound and lower bound for reference are useful for performance analysis during the evaluation of different settings. However, the details of model evaluation/test are not provided, for the global and personal models, for example, the setup and partitions of the datasets used in global or local test which are important in federated learning.**
>
> Thanks for this comment. These details can be found in:
>
> - The Appendix in Section A.11.3 “Federated Learning Experiments”.
>
> - The Experiments subsection in Section 4.4 in the main text of the paper (for CIFAR100), and
>
> - Appendix Section A.8 “Additional Few-shot Federated Learning Results” for Quickdraw.
>
> We realize that this spreads details across multiple locations, so to make it clearer, we updated the manuscript and summarized all the experimental details in section A.11.3 "Federated Learning Experiments" in the revised version of the paper.
>
> **In the first sentence of the last paragraph in Page 8, the authors mentioned that, no suitable federated learning system for comparison. As the key idea of the proposed method in the scenarios of federated learning is based on partial model sharing/update, why the authors did not include the comparison of some partial model transmission-based federated learning (FL) methods or in personalized FL (PFL) scenarios, e.g., the sparisification based FL and PFL in which the communication efficiency and the performance of personal models are investigated.**
>
> Our understanding is as follows: partial model sharing in federated learning considers situations where we have a shared backbone and personalized adapters and both the backbone and the adapters are learned (for example, see Pillutla et al., Federated Learning with Partial Model Personalization, Proceedings of the 39th International Conference on Machine Learning, PMLR 162:17716-17758, 2022). This is unlike our setup where the backbone is fixed (pretrained) and the adapters are shared across all users (i.e. they are not user-specific). For this reason, we do not compare to these methods. We have added a discussion of this related work to the paper (Section A.9 Connections to personalized federated learning in the updated manuscript).
>
> We are not aware of the sparsification based federated learning literature and could not find papers on this topic. Please could you provide a reference so we can comment on this?

---

> > ### Author Response · Authors · 2022-11-11
> > **Response to Reviewer BNuH (Part 2)**
> >
> > **In FL settings used on Tab. A.7 and Fig. A.3, the performance of conventional models based FL methods should be provided, with regard to accuracy and communication efficiency. It will help to demonstrate the effectiveness.**
> >
> > Thanks for the comment. We assume here that the reviewer is asking why we didn’t run the BiT+FedAvg baseline on the QuickDraw dataset, whereas we had for CIFAR100?
> > We didn’t include these experiments in the original version of our paper, as QuickDraw is not a common baseline to benchmark federated learning methods, and with our experiments on QuickDraw we wanted to show that FiT could be applied effectively to datasets that are not similar to natural image datasets that are used to pretrain backbones. We have now run these experiments and the results are as follows:
> >
> > |        | Parameter cost || 50 Clients ||| 100 Clients ||| 500 Clients|||
> > |---|---|---|---|---|---|---|---|---|---|---|---|
> > | Models | Per round | Overall| 2-Shot| 5-Shot | 10-Shot| 2-Shot   | 5-Shot   | 10-Shot| 2-Shot   | 5-Shot   | 10-Shot  |
> > | FiT   | **0.1M**       | **7M**  | 26.6±1.0     | 40.8±0.6 |  45.5±0.2 | 32.0±2.0     | 44.0±0.7 | 46.6±0.2 | 39.4±1.9    | 45.6±0.1 | 47.9±0.4 |
> > | BiT   | 242M           | 14.5B   | **37.5±2.2** | 42.4±1.7 | 44.4±0.7  | **40.4±0.5** | 44.2±1.4 | 46.4±0.9 | 41.1±2.5    | 45.7±2.6 | 47.5±1.7 |
> >
> > Table: Comparison of BiT and FiT in few-shot federated setting on Quickdraw for different numbers of clients and shots per client. The global setting is used. Both models were trained for 60 communication rounds. Parameter cost indicates the number of parameters sent in server-client communication. Per round is the number of parameters transmitted in each communication round and Overall is the number of parameters transmitted during the whole training.
> > We have also included these results in the revised version of the paper in Table A.8.
> >
> > For 5 and 10 shots per class, FiT performs similarly to BiT for all numbers of clients tested. However, for 2-shot, BiT outperforms FiT for all numbers of clients tested, but the performance gap is reduced as the number of clients is increased. We attribute this behavior to the use of the ProtoNets classifier. During training at each client’s optimization step, we construct a N-way local ProtoNets classifier using only one image per class, which may result in a classifier not robust enough for reasonable optimization of the FiLM layers. This hypothesis is also supported by Figure A.3, where we see a huge gap between the performance of Federated Learning and centralized learning (the upper bound) for 2-shot. In contrast, the linear head in BiT is not local and is trained using all clients’ data, thus avoiding this pitfall. This leads us to the following observation – if there is enough local client data to construct a robust metric-based classifier, then Naïve Bayes head helps to significantly reduce communication cost without sacrificing the final model quality. However, if there is not enough local client data, then the use of a linear classification head may be more appropriate.
> >
> > We thank the reviewer for raising this question and have included this comment in our manuscript.

---

> > > ### Author Response · Authors · 2022-11-11
> > > **Response to Reviewer BNuH (Part 3)**
> > >
> > > **In the writing, more discussion should be provided around the proposed method in the federated settings, such as the advantages and the difference from the conventional partial model based PFL methods.**
> > >
> > > Thanks for drawing our attention to partial model personalization in federated learning. As pointed out above, partial model based personalized federated learning is related to our personalized federated learning approach, but it is different in important regards. With regard to the similarities, the local Naïve Bayes head parameters in our approach may be considered as “personal” parameters (there are personalized parameters for each client), while the FiLM parameters could be viewed as “shared” parameters. However, there are a few major differences between the ideas:
> > > - The personalized parameters in our setting, i.e. the ProtoNets head, does not require an optimization loop to be learned. This simplifies deployment significantly.
> > > - The partial model personalization literature (e.g. see Pillutla et al., Federated Learning with Partial Model Personalization, Proceedings of the 39th International Conference on Machine Learning, PMLR 162:17716-17758, 2022; Collins et al., Exploiting shared representations for personalized federated learning, Proceedings of the 38th International Conference on Machine Learning, PMLR 139:2089-2099, 2021; ) is more concerned with proposing stable federated training algorithms that would work within heterogeneous settings, where clients have diverse data and standard FedAvg algorithm would fail, thus necessitating a need to introduce ‘personal’ parameters. In contrast, personalized heads are required in our setup as each user has a different classification task to perform. Moreover, we propose a particular architecture that would be highly suitable for federated learning applications, as the number of parameters required to be transmitted is small, while most methods in PFL propose architecture agnostic optimization algorithms. Also, our approach can then be trained with an arbitrary Federated Learning algorithm.
> > > - Another distinctive difference between the model personalization federated learning literature and our work is that in the former most of the methods train all model parameters, while in our work we used a deep pretrained network and fine-tune only the FiLM layers.

---

> > > > ### Comment · Reviewer_BNuH · 2022-11-22
> > > > **Thanks for the additional clarifications and experiments.**
> > > >
> > > > The authors have addressed most of my concerns.  I am happy to increase my score to 6, leaning towards accept. As for the sparsification based federated learning literatures, some of them could be:
> > > > https://proceedings.mlr.press/v162/dai22b/dai22b.pdf
> > > > https://dl.acm.org/doi/abs/10.1145/3485730.3485929
> > > > https://arxiv.org/abs/2112.09824

---

> > > > > ### Author Response · Authors · 2022-11-23
> > > > > **Comment on Sparse Federated Learning References**
> > > > >
> > > > > We thank reviewer for providing the references [1-3] on sparse federated learning, and want to highlight the main differences between our work and this line of research:
> > > > > - All the papers that you listed on sparse federated training are mainly trying to optimize for a good binary mask (either global or personalized) to determine which network weights should be used for inference. In contrast, we are using all the model weights for inference, while updating only the FiLM adapter parameters during training and aggregation.
> > > > > - None of the references are leveraging large pretrained models, whereas our work builds on the extremely good performance of these.
> > > > > - While not directly comparable, on CIFAR100, our classification accuracy is higher and our communication costs are lower compared to those reported in [1] and [3].
> > > > >
> > > > > [1] https://proceedings.mlr.press/v162/dai22b/dai22b.pdf
> > > > >
> > > > > [2] https://dl.acm.org/doi/abs/10.1145/3485730.3485929
> > > > >
> > > > > [3] https://arxiv.org/abs/2112.09824

---

### Official Review · Reviewer_upHx · 2022-11-03

**Confidence:** 2
**Correctness:** 4
**Technical Novelty And Significance:** 3
**Empirical Novelty And Significance:** 3
**Recommendation:** 8

**Clarity, Quality, Novelty And Reproducibility:**

The paper is clearly written and easy to follow. The work is original, combining previously existing ideas from the transfer learning and meta-learning communities.

There are some minor typos in the paper, such as in the bottom part of the second page "... while using using ≈ 1% of the updateable parameters when compared to the leading transfer learning method BiT".


**Strength And Weaknesses:**

Strengths: The paper introduces a novel strategy to combine ideas from transfer learning and meta-learning communities for developing data-efficient and parameter-efficient learning systems. The authors motivated the proposed method well. The method achieved superior performance at low-shot on several benchmarking datasets. In addition, the authors further demonstrated FIT's parameter efficiency and superior accuracy in distributed low-shot personalization and federated learning applications. The evaluation is relatively convincing.

Weaknesses: The paper includes a range of quantitative evaluation results, but it would be better to also provide a more in-depth explanation of the experimental results and the authors' insights on why their proposed method works, and in which settings it would suffer.

**Summary Of The Paper:**

This paper proposes a novel idea to improve few-shot learning by synergizing ideas from the transfer learning and meta-learning communities. Specifically, from transfer learning, it leverages the backbones pretrained on large image datasets and fine-tuned parameter-efficient adapters, while from meta-learning, it uses metric learning-based final layer classifiers trained with episodic protocols. The authors conducted experiments on several downstream datasets, illustrating that their proposed method, FIT, achieved superior accuracy at low-shot on benchmarking datasets. In addition, they demonstrated the method's parameter efficiency in distributed few-shot learning including model personalization and federated learning tasks.

**Summary Of The Review:**

This paper proposes a novel method combining existing ideas from transfer learning and meta-learning, which enables a parameter and data-efficient network architecture for low-shot learning. The authors conducted a range of experiments on standard datasets to illustrate the effectiveness of their proposed method. This strategy could be inspiring to the community and potentially generalize beyond imaging classification tasks.

---

> ### Author Response · Authors · 2022-11-11
> **Response to Reviewer upHx**
>
> Thanks for taking the time to review our paper. Responses to your questions and issues are below:
>
> **The paper includes a range of quantitative evaluation results, but it would be better to also provide a more in-depth explanation of the experimental results and the authors' insights on why their proposed method works, and in which settings it would suffer.**
>
> FiLM works well for low shots per class because it has a small number of parameters to adapt – this gets you a long way with large pre-trained backbones -- and it also prevents over-fitting as compared to adapting all the parameters in the backbone. Evidence: eventually as dataset size increases, full body adaptation wins / is as good as FiLM. For larger models, this transition happens at a larger number of data points and so we expect adapters such as FiLM to be especially useful as the field transitions to using large foundation models. Figure 3 provides an empirical justification of this argument, showing that FiT-LDA outperforms BiT at low-shot.
>
> The linear head is more flexible and works best of all methods in high data, but in the low- to medium-shot setting the linear head appears to lead to increased overfitting and the meta-trained LDA head performs best in this region.
>
> The approach suffers when there are a fairly large number of data points and the dataset is very far from the data it has been pretrained on (ImageNet) e.g. see the DSprites results in Table 3 where BiT beats FiT-LDA by a large margin. In addition, the LDA and QDA variants of the Naïve Bayes head are more computationally expensive compared to a linear head since LDA and QDA require inverting $d_b \times d_b$ matrices for each training iteration.
>
> We have added this information to Section A.6 Discussion of the empirical results of the revised version of the paper.
>
> **There are some minor typos in the paper, such as in the bottom part of the second page "... while using using ≈ 1% of the updateable parameters when compared to the leading transfer learning method BiT".**
>
> Thanks for spotting this, we have fixed this in the updated version of the paper.

---

### Author Response · Authors · 2022-11-11
**Thanks to all the reviewers!**

We thank the reviewers for their thoughtful reviews. Four reviewers are recommending accepting the paper (two strongly and two weakly), and one is recommending rejection (weakly).

The reviewers agreed that:

i) our work is well-motivated (“The paper is **well motivated**...” (Reviewer DFdq), “**The motivation is well-elaborated**...” (Reviewer BNuH), “The authors **motivated the proposed method well**.” (Reviewer upHx))

ii) our work has provided an extensive set of experiments supporting the efficiency and effectiveness of the proposed method (“**The adequate experimental results**” (Reviewer 8TEt), “The paper is well-organized **with detailed experiments and comparisons**” (Reviewer YJfE), “The paper provides **sufficient and convincing experiment results**” (Reviewer DFdq), “**The evaluation is relatively convincing**” (Reviewer upHx))

The reviewers requested various clarifications about the proposed method and asked for a better understanding of why the method works better than standard approaches to transfer learning.

We have updated the paper based on the reviewer feedback, improving the readability, adding explanations, clarifications, and fixing typos. We have also made efforts to connect the work more concretely to the personalized federated learning literature. We also ran an additional baseline on the federated learning experiments as sensibly requested by reviewer BNuH. An updated version of the paper draft has now been uploaded with changes marked in red.

---

### Decision · Program_Chairs · 2023-01-20

**Decision:**

Accept: poster

**Justification For Why Not Higher Score:**

While the problem is well-motivated and the experiments confirm the effectiveness of the proposed method, the proposed approach is mainly combination of existing ideas in transfer learning and meta-learning.

**Justification For Why Not Lower Score:**

The proposed method is effective and of interest to the ICLR community.

**Metareview: Summary, Strengths And Weaknesses:**

The paper proposes FiLM Transfer (FiT) which combines ideas from transfer learning and meta-learning for developing data-efficient and parameter-efficient learning. Empirical results shows that FiML achieves superior performance at low-shot on several benchmarking datasets. The results also confirm FIT's parameter efficiency and superior accuracy in distributed low-shot personalization and federated learning applications. The paper is well motivated and the works provides an extensive set of experiments supporting the efficiency and effectiveness of the proposed method. Thus, it is of value to the ICLR community.

**Note From Pc:**

if the above contains the word "oral" or "spotlight" please see: "oral" presentation means -> notable-top-5% and "spotlight" means -> notable-top-25%. As stated in our emails, we are disassociating presentation type from AC recommendations